

# The Impact of Upwelling on the Intensification of Anticyclonic Ocean Eddies in the Caribbean Sea

Carine G. van der Boog[1], Julie D. Pietrzak[1], Henk A. Dijkstra[2], Nils Brüggemann[5], René M. van Westen[2], Rebecca K. James[3], Tjeerd J. Bouma[3], Riccardo E.M. Riva[4], D. Cornelis Slobbe[4], Roland Klees[4], Marcel Zijlema[1], and Caroline A. Katsman[1]

[1]Environmental Fluid Mechanics, Civil Engineering and Geosciences, Delft University of Technology, Stevinweg 1, 2628 CN Delft, the Netherlands
[2]Institute for Marine and Atmospheric Research, Utrecht University, Princetonplein 5, 3584 CC Utrecht, the Netherlands
[3]Estuarine and Delta Systems, Royal Netherlands Institute for Sea Research and Utrecht University, Korringaweg 7, 4401 NT Yerseke, the Netherlands
[4]Geoscience and Remote Sensing, Delft University of Technology, Stevinweg 1, 2628 CN Delft, the Netherlands
[5]University of Hamburg, Bundesstrasse 53, 20146 Hamburg, Germany

**Correspondence:** Carine van der Boog, c.g.vanderboog@tudelft.nl

**Abstract.** The mesoscale variability in the Caribbean Sea is dominated by anticyclonic eddies that are formed in the eastern part of the basin. These anticyclones intensify on their path westward while they pass the coastal upwelling region along the Venezuelan and Colombian coast. In this study, we used a regional model to show that this westward intensification of Caribbean anticyclones is driven by the advection of cold upwelling filaments. These dense filaments are advected by the

anticyclones, leading to an increase of the horizontal density gradients at the western side of the anticyclones. Following the thermal wind balance, the increased density gradients result in an increase of the vertical shear of the anticyclones and to their westward intensification. To assess the impact of variations in upwelling on the anticyclones, several simulations were performed in which the northward Ekman transport (and thus the upwelling strength) is altered. As expected, stronger (weaker) upwelling is associated with more stronger (weaker) offshore cooling and a more (less) westward intensification of

the anticyclones. The simulations with weaker upwelling show farther advection of the Amazon and Orinoco River plumes into the basin. The dispersion of the river plumes affects the formation process of the anticyclones, where the horizontal density gradients were mainly determined by the salinity gradients of the river plume and not by temperature gradients that were associated with upwelling.

## 1  Introduction

The circulation in the Caribbean Sea is characterized by the throughflow of the wind-driven subtropical North-Atlantic gyre (Gordon, 1967), known as the Caribbean Current. This flow is part of the upper branch of the Meridional Overturning Circulation (Johns et al., 2002), and is highly variable, which manifests itself in meanders and mesoscale eddies. The temporal and spatial variability of the Caribbean Current is influenced by regional differences in wind (Nystuen and Andrade, 1993; Andrade and Barton, 2000; Chang and Oey, 2013; Lems-de Jong, 2017), fresh water inflow from the Amazon and Orinoco River plumes



(Chérubin and Richardson, 2007; Beier et al., 2017), resonating Rossby waves (Hughes et al., 2016) and, at interannual time scales, by the El Niño Southern Oscillation (Alvera-Azcárate et al., 2009; Beier et al., 2017).

Surface drifter data show that the mesoscale variability in this region is dominated by anticyclonic eddies (Molinari et al., 1981; Centurioni and Niiler, 2003; Richardson, 2005). These anticyclones are formed in the eastern part of the basin and
transport low salinity anomalies originating from the Amazon and Orinoco river plumes westward (Silander, 2005; Rudzin et al., 2017; van der Boog et al., 2019). They can be generated from the interaction of the flow with the topography (Molinari et al., 1981), from the meandering current (Andrade and Barton, 2000), from instabilities due to the presence of the river plume (Chérubin and Richardson, 2007), and from perturbations caused by the interaction of North Brazil Current Rings (NBC Rings) with topography (e.g., Simmons and Nof, 2002; Goni and Johns, 2003; Jochumsen et al., 2010). Jouanno et al. (2009) used a
model to clarify the dominant generation mechanism, and concluded that the mean flow in the Caribbean Sea is intrinsically unstable, which means that any perturbation could trigger the formation of Caribbean anticyclones.

After formation, the Caribbean anticyclones are advected westward with the mean flow (Gordon, 1967; Andrade and Barton, 2000), and propagate along the wind-driven upwelling regions along the South-American coast. Based on a hydrographic time series of the upwelling in Cariaco Basin, Astor et al. (2003) found that the interannual variability of temperatures in the
upwelling region is affected by the anticyclones that advect cold filaments of upwelled waters offshore. This advection affects the ecosystem as it transports larvae and nutrients offshore (Andrade and Barton, 2005; Baums et al., 2006). The advection of these cold filaments also leads to cooling of the interior of the Caribbean Sea (Jouanno and Sheinbaum, 2013). The anticyclones leave the Caribbean Sea through Yucatan Channel. Here, model studies have shown that Caribbean anticyclones could influence eddy-shedding events of the Loop Current (Oey et al., 2003; Murphy et al., 1999; Carton and Chao, 1999; Candela, 2003; van
Westen et al., 2018).

During their propagation, the anticyclones become more energetic (Carton and Chao, 1999; Pauluhn and Chao, 1999; Andrade and Barton, 2000; Richardson, 2005). Although this intensification is clearly present in observations (Carton and Chao, 1999; Pauluhn and Chao, 1999; Andrade and Barton, 2000; Richardson, 2005), only a few studies elaborate (briefly) upon the dynamics of this intensification. Based on surface drifter data, Richardson (2005) suggested that the anticyclonic shear of the
Caribbean Current could amplify the anticyclones. In contrast, Andrade and Barton (2000) found, based on satellite altimetry data, a direct relationship between the maximum curl of the wind stress and the westward intensification of anticyclones. This relationship highlights that wind stress alters the life cycle of the anticyclones. Jouanno et al. (2009) used a regional model to study the life cycle of Caribbean anticyclones and computed the mechanical energy balance of the flow in this region. Although this balance shows that baroclinic instabilities provide the energy necessary for the westward growth of the anticyclones, it does
not explain what drives the westward intensification of the anticyclones.

In this study, we hypothesize that this intensification is driven by the offshore advection of cold upwelling filaments that cool the interior of the basin. The advection of the filaments results in denser surface waters in the western part of the basin compared to the eastern part of the basin, suggesting that the density difference between the relatively light anticyclones and the surrounding waters will increase. Following the thermal wind balance, it can be expected that the vertical shear of the
anticyclone will increase, and consequently its strength.



To test this hypothesis, we use a regional model in which we vary the upwelling strength. The coastal upwelling is adjusted by altering the magnitude of the zonal wind stress with a constant. The curl of the wind stress is kept constant by applying the adjustment over the full model domain. A description of the model is provided in Section 2, and is followed by a comparison between the modeled flow and observations (Section 3). Section 4 and 5 contain the analysis of the westward intensification

of the anticyclone and how this is related to the advection of upwelling filaments. The sensitivity of the mean flow and eddy variability to changes in upwelling strength are discussed in Section 6.

## 2   Model configuration and methods

### 2.1   Model configuration

The numerical simulations were performed with the hydrostatic configuration of the Massachusetts Institute of Technology

(MIT) primitive equation model (Marshall et al., 1997). The computational domain extended from $99^o$W to $55^o$W and from $6^o$N to $33^o$N (Fig. 1a), and was set up with a horizontal resolution of $1/12^o$, which is well below the internal Rossby radius of deformation in this region (60-80 km, Chelton et al., 1998). In the vertical, the model contained 50 levels in z-coordinates, increasing in depth from 1 m at the surface towards 459 m at the lower levels. The time step was 240 s. All simulations had a total duration of 25 years, of which the first five years were considered as spin-up and were excluded from the analysis. The

model output was saved as 5-day averaged fields. The simulations were initialized with time-averaged fields for sea-surface height, temperature, salinity and velocity. These fields were obtained from the years 2007-2017 of the Operational Mercator global ocean analysis (Mercator) of the E.U. Copernicus Marine Service Information. The vertical diffusion of tracers was parameterized with the GGL90 mixed layer parameterization (Gaspar et al., 1990). The horizontal diffusion was parameterized with the Redi-scheme (diffusion coefficient of 125 m$^2$ s$^{-1}$, Redi, 1982). The sub-grid scale mixing was parameterized with

Smagorinsky viscosities (Smagorinsky et al., 1993).

A sponge layer with a meridional width of $1.25^o$ (15 grid cells) was applied at the open boundaries that relaxed the velocity fields towards the time-averaged fields of Mercator (years 2007-2017) with a relaxation time that varied linearly from a month to a day at the boundary. At these boundaries, temperature, salinity, and velocities in zonal and meridional directions were prescribed. At the surface, a fresh-water flux, temperature restoring and a wind stress were applied (Fig. 1b-d). The temperature

was restored towards averaged sea-surface temperature (SST) of Mercator with a relaxation time scale of one month. The surface fresh-water flux was obtained from the diagnosed fresh-water flux of a 250-year simulation described in Le Bars et al. (2016). The Orinoco River, Magdalena River and Mississippi River are prescribed as stationary fresh water fluxes at the open boundaries with discharges based on Fekete et al. (2000).

The most realistic simulation, referenced to as Ekman100, contained stationary boundary conditions obtained from the years

2007-2017 of Mercator (Fig. 1). In this simulation, the northward Ekman transport at the upwelling regions corresponds to of the year-averaged northward Ekman transport (100%), which was computed from the wind fields of ERA-Interim (Dee et al., 2011). Prescribing stationary values implied that the NBC Rings, which can trigger the formation of the anticyclones, were not represented at the boundaries. However, Jouanno et al. (2009) and Lin et al. (2012) showed that a realistic eddy field in the



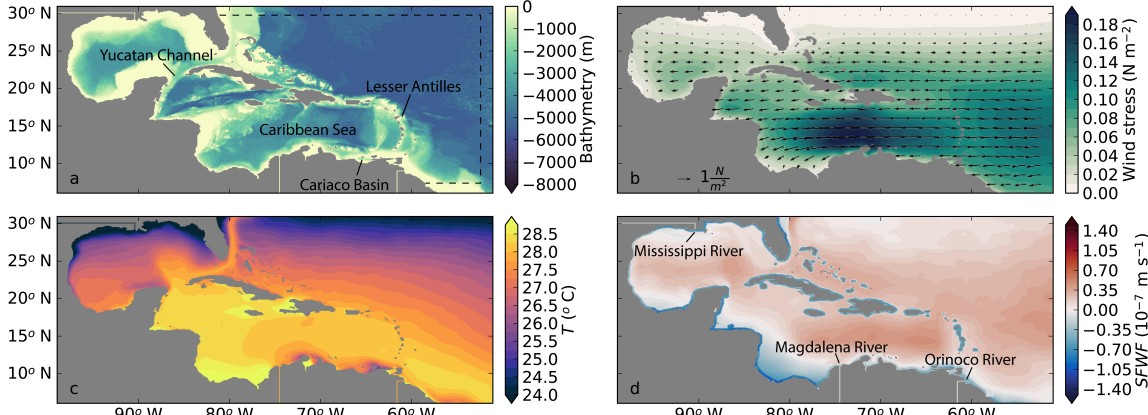

**Figure 1.** Setup of the regional model in MITgcm. (a) Bathymetry in m. The sponge layer is indicated with the dashed line. (b) Magnitude of the wind stress. The direction in indicated with the vectors. Not all velocity vectors are shown for clarity purposes. (c) Surface temperature field in $^oC$ used for restoring. (d) Surface fresh water flux in m s$^{-1}$. Positive values correspond to a net surface buoyancy loss.

Caribbean Sea can be obtained without the presence of NBC Rings. Moreover, in Section 3 we show that a realistic eddy field is obtained with these boundary conditions. Therefore, we considered these boundary conditions sufficient for the purpose of this study.

To investigate the effect of wind-driven upwelling on the westward intensification of anticyclones, the zonal wind forcing ($\tau_x$) was altered between simulations with a constant proportional to the wind stress at the upwelling regions (Fig. 2). This procedure ensured we only change the upwelling strength and not the curl of the wind stress. In our reference simulation, Ekman100, the constant was set to zero. In Ekman150, the coastal zonal wind stress was 50% stronger than the wind stress in Ekman100, resulting in a theoretical increase of the northward Ekman transport of 50%. The wind stress along the coast was weaker in the Ekman50, leading to a theoretical weaker upwelling (50%) in this simulation compared to Ekman100. The same principle was applied in Ekman75 and Ekman125.

The adjustment of the zonal wind forcing resulted in changes in mesoscale variability over the full domain. Because the applied constant was optimized to alter the northward Ekman transport in the Cariaco and Guajira upwelling regions, it resulted in unrealistic magnitudes of the zonal wind stress (and thus of the northward Ekman transport) in some other regions. Therefore, we will only analyze the impact of the changes in wind forcing on the mesoscale variability close to the Cariaco and Guajira upwelling regions, and disregard other coastal upwelling regions.

The range of $\tau_x$ applied in the simulations covers the seasonal variations (Fig. 2). The magnitude of the zonal wind stress in Ekman50 was similar to the observed zonal wind stress in fall, while the stronger zonal wind stress in Ekman125 was comparable to the wind stress during winter and summer months. The zonal wind stress in Ekman150 was stronger compared to the seasonal cycle, but is similar to zonal wind stresses that are observed during years with anomalously high wind velocities





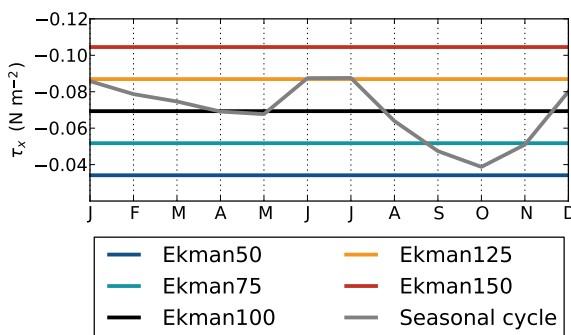

**Figure 2.** Average zonal wind stress in N m$^{-2}$ in the southern Caribbean Sea (10.5$^o$N-12.5$^o$N, 65$^o$W-70$^o$W) in the different simulations. The seasonal variation of the zonal wind stress at this location, obtained from ERA-Interim (Dee et al., 2011), is shown for reference. Negative values correspond to easterly winds.

(Whyte et al., 2008). We will use this set of simulations with the different to study aspects of the seasonal and interannual variability of Caribbean anticyclones.

## 2.2 Methods

To analyze the behavior of the mesoscale eddies, we analyzed both the eddy kinetic energy (EKE) and the individual eddies.

Following Jouanno et al. (2012), we calculated the EKE from 5-day averaged velocity fields that were high-pass filtered with a 125-day running mean. This 125-day period was long enough to capture the variability of the eddies, which have a characteristic period of 50-100 days (Jouanno et al., 2008), but was shorter than the interannual variability.

To gain insight into the westward intensification of Caribbean anticyclones, we used the py-eddy tracker to follow them (Mason et al., 2014). The py-eddy tracker used 5-day averaged sea-level anomaly fields to identify near-circular anomalies.

Anomalies were computed with respect to 20-year averaged fields. Negative anomalies were identified as cyclones, and positive anomalies as anticyclones. More detailed information about the numerics of the py-eddy tracker can be found in Mason et al. (2014). We set a minimum life time of 75 days. Taking into account the average westward propagation velocity of 0.13 m s$^{-1}$ (Richardson, 2005), the anticyclones propagate approximately 840 km during this minimum life time.

At the locations of the eddies provided by the eddy tracker, we extracted the amplitude ($A_{eddy}$), swirl velocity ($u_{swirl}$),

radius ($R_{eddy}$) and properties from the model output to assess their characteristics. The amplitude was defined as sea-surface height difference between the eddy and the 20-year average sea-surface height at that location, the swirl velocity as the average of the maximum northward and maximum southward velocity of the eddy. The radius ($R_{eddy}$) was defined as half the distance between the location of these velocities. The properties ($T, S, \sigma$) of the eddies and background, taken at the core of the eddies, were averaged over the upper 50 m of the water column to limit the influence of the surface forcing on the properties. Note that

the temperature restoring and surface fresh-water flux influence the surface layer only. Here, the background properties were





defined as the 125-day averaged values of the upper 50 m at the location of the eddy obtained from the py-eddy tracker. The differences between the properties of the eddies and the background are computed as $\Delta = eddy - background$.

To understand how the thermal wind balance affected the westward intensification of the anticyclones in each simulation, we computed the strength of the horizontal density gradients ($|\nabla\sigma|$). These gradients were calculated over the upper 50 m of the water column for 5-day averaged density fields as follows:

$$|\nabla\sigma| = \sqrt{\left(\frac{\partial\sigma}{\partial x}\right)^2 + \left(\frac{\partial\sigma}{\partial y}\right)^2}, \tag{1}$$

where $\sigma$ is the density. The contribution of temperature ($|\nabla\sigma_T|$) and salinity ($|\nabla\sigma_S|$) to the horizontal density gradients was computed in a similar manner, where the density part was calculated as $\sigma_T = \rho_0\alpha T$ and $\sigma_S = \rho_0\beta S$, respectively. Here, $\alpha$, $\beta$ and $\rho_0$ are constants; $\alpha = -3.1 \times 10^{-4}\ ^oC^{-1}$, $\beta = 7.2 \times 10^{-4}$ psu$^{-1}$, and $\rho_0 = 999.8$ kg m$^{-3}$.

The eddy kinetic energy and the density gradients of the anticyclones are estimated by considering only the EKE and density gradients around the core of an eddy at each 5-day averaged field. The considered region was limited to $1.5 \times R_{eddy}$. This restriction was applied to ensure that the advection of cold filaments and other mesoscale variability was excluded from the analysis.

## 3 Performance of the regional model

### 3.1 Mean flow

The modeled surface current in Ekman100 enters the Caribbean Sea through the southern passages of the Lesser Antilles (Fig. 3a), which is similar to what is seen in observations (Johns et al., 2002). In line with surface drifter data (Centurioni and Niiler, 2003), the largest velocities and majority of the westward transport is located south of 14$^o$N (Fig. 3a). Further westward, the modeled flow accelerates over shallow topography at 17$^o$N, where it continues northwestward towards Yucatan Channel into the Gulf of Mexico. The model has a mean transport of 24.9 Sv through Yucatan Channel, which is similar to the 23.8 Sv that was observed by Sheinbaum et al. (2002).

In the southwest of the Caribbean Sea, the modeled surface currents displays a cyclonic recirculation (Fig. 3a). This recirculation is known as the Panama-Colombia Gyre (PCG) and is wind-driven (Centurioni and Niiler, 2003; Andrade, 2003). Part of the PCG continues as an eastward subsurface countercurrent, which results from the Sverdrup circulation in the North Atlantic tropical cell (Andrade, 2003). The countercurrent flows below the upwelling region, and its depth is related to the wind strength (Andrade, 2003; Andrade and Barton, 2005). The model is able to reproduce the subsurface countercurrent at a depth of approximately 100 m, which is slightly higher in the water column than at the 200 m observed by Andrade (2003) and the 150 m observed by Hernandez-Guerra and Joyce (2000); the modeled strength (0.11 m s$^{-1}$) is comparable to their observations.

Ekman100 displays a strong meridional density gradient that varies between $\sigma = 25.1$ kg m$^{-3}$ in the south (11$^o$N) and $\sigma = 22.7$ kg m$^{-3}$ in the north (18$^o$N, Fig. 3b). The strongest meridional gradients are co-located with the Caribbean Current.





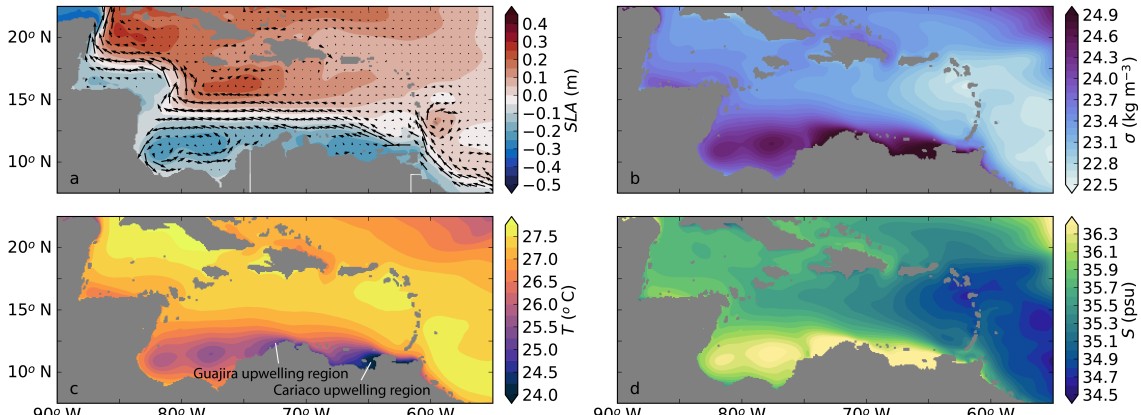

**Figure 3.** 20-year near-surface average properties of the Caribbean Sea in Ekman100. (a) Mean sea-level anomaly (SLA) in m with velocity vectors. Surface current vectors are only shown every eighth grid cells for clarity purposes. (b) Density ($\sigma = \rho - 1000$) in kg m$^{-3}$, (c) Temperature in $^o$C and (d) Salinity in psu. Properties in panel b-c-d are averages over the upper 50 m.

The location of the dense waters coincide with the two major upwelling regions: Cariaco and Guajira. The upwelled waters are colder and more saline than the surrounding surface waters (Fig. 3c,d).

The Cariaco upwelling region in the southeast of the Caribbean Sea is located at 63$^o$W-65$^o$W, 10$^o$N-12.5$^o$N (Rueda-Roa and Muller-Karger, 2013). The modeled minimum (24.9 $^o$C) and maximum temperatures (27.4 $^o$C) in Cariaco Basin are

less extreme than the observed temperatures (minimum: 20.3 $^o$C, maximum: 30.1 $^o$C, Rueda-Roa and Muller-Karger, 2013). Although the model is not able to capture this seasonal variability, the modeled average SST (25.4 $^o$C) is similar to observations (25.2 $^o$C, Rueda-Roa and Muller-Karger, 2013). Note that these modeled temperatures are lower than the restoring temperature (Fig. 1c), which indicates that the model is able to reproduce coastal upwelling. The Guajira upwelling region is located west of the Cariaco upwelling region, between 69$^o$W and 74$^o$W (Rueda-Roa and Muller-Karger, 2013). Here, the observed average

SST is slightly higher (25.5 $^o$C) than in the Cariaco upwelling region (Rueda-Roa and Muller-Karger, 2013). The model displays a similar temperature difference between the two upwelling regions (26.1 $^o$C in Guajira, Fig. 3d).

In addition to the meridional density gradient, the model also displays a clear zonal density gradient in the Caribbean Sea (Fig. 3b): the Caribbean Current is relatively light as it enters the Caribbean Sea. This zonal density gradient is mainly due to a zonal salinity gradient, and to a lesser extent to a zonal temperature gradient (Fig. 3c-d). As mentioned in the introduction, the

zonal temperature and salinity gradients are related to the offshore advection of cold and saline upwelling filaments (Jouanno and Sheinbaum, 2013). According to observations, the zonal salinity gradient is also related to the dispersal of the Amazon and Orinoco River plumes into the basin (e.g., Hu et al., 2004; Chérubin and Richardson, 2007). In the model, the Amazon River plume enters the domain at the southern boundary, while the fresh water of the Orinoco River enters the domain at 61$^o$W, 9$^o$N (Fig. 1d). The river plume is advected with the mean flow and becomes more saline through mixing with the saline (upwelled)

surface waters in the basin, and through evaporation. Overall, the zonal and meridional gradients of salinity and temperature





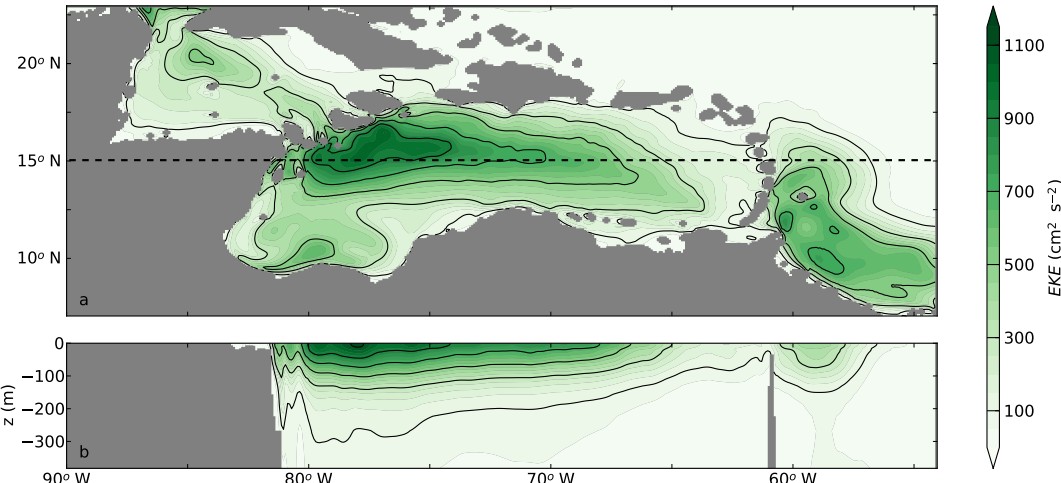

**Figure 4.** Eddy kinetic energy (EKE) in cm$^2$ s$^{-2}$ of Ekman100 obtained from 5-day averaged velocity fields that are high-passed filtered with a 125-day mean. (a) 20-year average of the EKE in the upper 50 m of the water column and (b) cross section of EKE at 15$^o$N over the upper 400 m.

are similar to observations, indicating that the model is able to capture the upwelling, the advection of the river plume water and the relevant mesoscale processes.

## 3.2 Eddy kinetic energy

In line with observations (Richardson, 2005; Carton and Chao, 1999), we find that the flow in the Caribbean Sea is highly

variable (Fig. 4). In the eastern part of the basin, the surface EKE is relatively low (100-300 cm$^2$ s$^{-2}$, Fig 4a). The EKE increases westward towards a maximum >900 cm$^2$ s$^{-2}$ at 78$^o$W. The modeled magnitude of EKE is higher than found in satellite altimetry (>600 cm$^2$ s$^{-2}$, Jouanno et al., 2012), but it is more similar to estimates obtained from surface drifters (>900 cm$^2$ s$^{-2}$, Richardson, 2005). This is in line with other modeling studies (Jouanno et al., 2008, 2012), and this discrepancy is mainly attributed to the coarse resolution (0.25$^o$) of the gridded altimetry data products (Jouanno et al., 2008). The modeled

spatial variability of EKE corresponds well to satellite altimetry (Jouanno et al., 2012; Ducet et al., 2000). In line with the results of Jouanno et al. (2008), the magnitude of EKE at depth increases towards the west (Fig. 4b).

The variability of the Caribbean Sea manifests itself in the presence of mesoscale eddies (Fig. 5). In line with observations (Centurioni and Niiler, 2003), the mesoscale eddies are predominantly anticyclonic (Fig. 5a). As expected, the surface densities of these anticyclones are lighter than those of the surrounding surface waters (Fig. 5b). The density differences are due to both

temperature and salinity (Fig. 5c,d). Figure 5c also displays the northward advection of a cold filament. Similar filaments have been observed, and it is known that these can be advected several hundreds of kilometers from the upwelling region (Andrade and Barton, 2005). This advection results in the offshore cooling of surface waters (Jouanno and Sheinbaum, 2013).

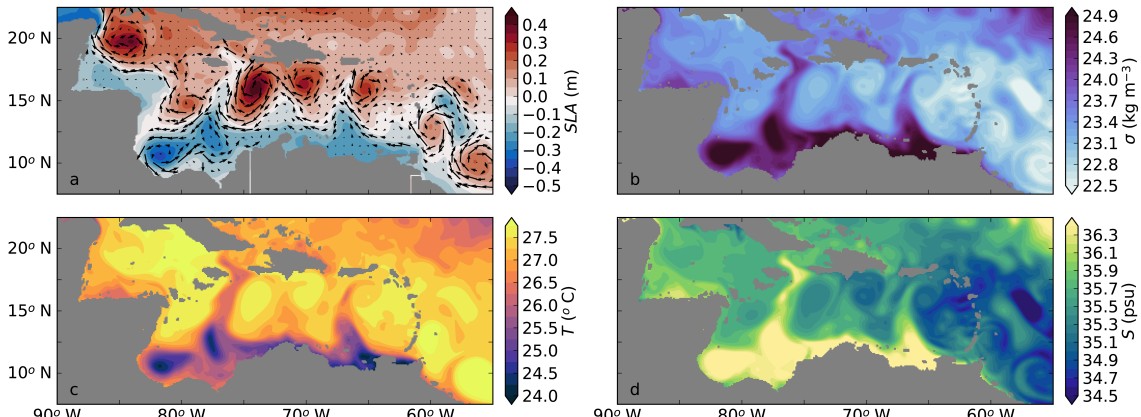

**Figure 5.** Near-surface properties of the Caribbean Sea, averaged over 5-days in Ekman100 in year 20; (a) Mean sea-level anomaly (SLA, in m) with velocity vectors. Surface current vectors are only shown every eighth grid cells for clarity purposes. (b) Density ($\sigma = \rho - 1000$) in kg m$^{-3}$, (c) Temperature in $^o$C and (d) salinity in psu. Properties in panels are averages over the upper 50 m.

### 3.3 Eddy characteristics

From the py-eddy tracker (Mason et al., 2014), we found that on average, 8.55 anticyclones and 5.70 cyclones are formed in the reference simulation Ekman100 east of 75$^o$W, between 12.5$^o$N-17.5$^o$N every year. These numbers are in line with the 8-12 anticyclones per year estimated from surface drifter data (Richardson, 2005). To our knowledge, there are no previous

estimates for the formation rate of cyclones in the Caribbean Sea. The mesoscale eddies are close to geostrophic balance, with an average Rossby number of 0.15±0.01 (anticyclones) and 0.14±0.03 (cyclones), where the Rossby number is calculated as $Ro = u_{swirl}/(R_{eddy} * f)$, in which $u_{swirl}$ is the swirl velocity of the eddy, $R_{eddy}$ is the radius and $f$ the Coriolis parameter.

The anticyclones have an average amplitude of $A_{eddy} = 0.17$ m and swirl velocities of $u_{swirl} = 0.60$ m s$^{-1}$ between 65$^o$W-75$^o$W and 12.5$^o$N-17.5$^o$N. This amplitude and swirl velocity are similar to those found in hydrographic surveys (Silander,

2005; Rudzin et al., 2017; van der Boog et al., 2019), and similar to velocities obtained from surface drifters ($u_{swirl} = 0.5$ m s$^{-1}$, Richardson, 2005). In general, the core of the anticyclones is warmer ($\Delta T = +0.15$ $^o$C) and fresher ($\Delta S = -0.18$ psu) than the surrounding surface waters (Fig. 5c-d).

The cyclones are less energetic than the anticyclones, and have an average amplitude of $A_{eddy} = -0.16$ m and swirl velocity of $u_{swirl} = 0.50$ m s$^{-1}$. Observations indicate that cyclones are generated near topographic features in the Caribbean Sea

(Richardson, 2005), and can contain upwelling waters (Andrade and Barton, 2005). The modeled properties of the cyclones are in agreement with these observations.

A strong and significant correlation between the amplitude of the tracked mesoscale eddies and the surface EKE confirms that the westward growth of EKE is related to the strength of the eddies. We find that 57 of the anticyclones that are formed in the Caribbean Sea during the 20 years of the simulation propagate from a region with weak EKE ($<65^o$W) towards a region



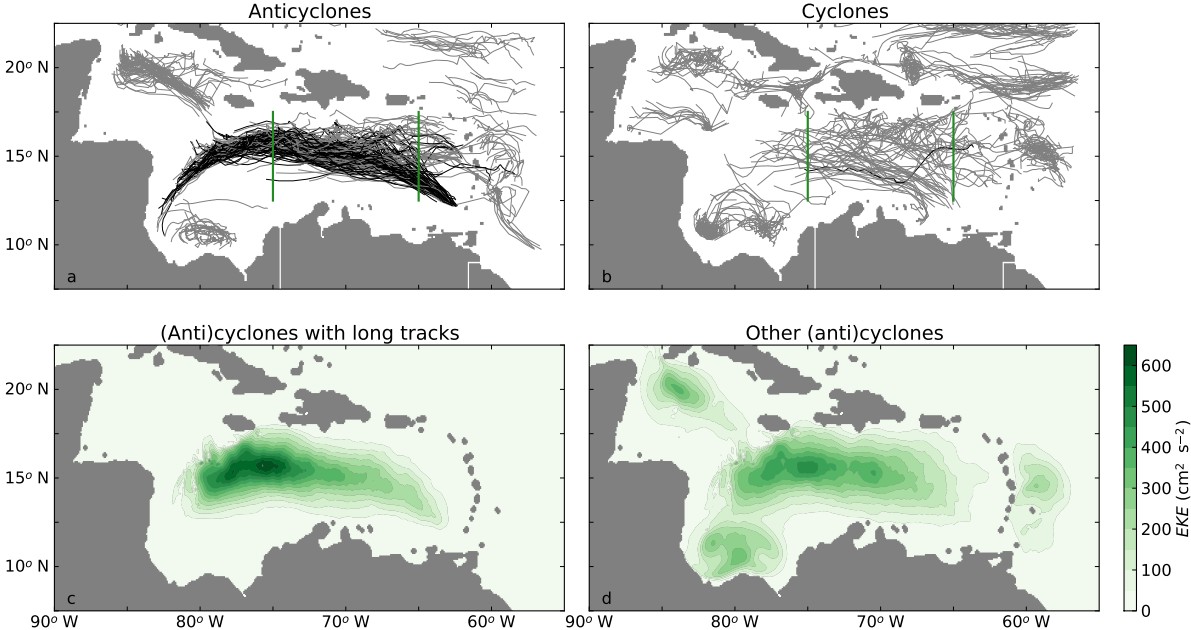

**Figure 6.** Paths of (a) anticyclones, and (b) cyclones that are identified by the py-eddy tracker (Mason et al., 2014) in Ekman100 during 20 years of model simulation. The black lines indicate tracks that pass both cross-sections at $65^o$W and $75^o$W (green lines). All other tracks are indicated in grey. (c) Estimate of EKE in cm$^2$ s$^{-2}$ of the long tracks (black lines in panels a and b). (d) Estimate of EKE of all other tracks (grey lines in panels a and b).

with high EKE ($>75^o$W). The paths of these anticyclones are indicated with the black lines in Figure 6a. The other 96 of the anticyclones that are formed in the Caribbean Sea during the 20 years of the simulation are either generated west of $65^o$W or do not pass $75^o$W (grey lines in Fig. 6a). In contrast to the anticyclones, cyclones display relatively short tracks: Only one cyclone passes both $65^o$W and $75^o$W (black line in Fig. 6b). This is similar to observations of Richardson (2005) who showed

5    that the cyclones follow a different path than anticyclones.

To assess the contribution of the anticyclones with the long tracks compared to the total EKE variability, we calculated their EKE by taking into account the EKE within $1.5 \times R_{eddy}$ of each eddy as described in Section 2. The spatial distribution of EKE due to these eddies (Fig. 6c) is similar to the spatial distribution of the total EKE (Fig. 4a). Notably, the magnitude of the EKE of these selected eddies is approximately 55% of the total EKE (Fig. 4), while it is computed from only 37% of

10   the eddies (57/153 anticyclones in 20 years). This shows that the modeled westward increase of EKE is dominated by the westward increase of EKE of a small number of long-lived eddies, which is also confirmed by the weaker eddy kinetic energy of the eddies with shorter tracks (Fig. 6d). Because our results shows that the eddies with long tracks, which are predominantly

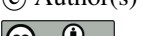



anticyclonic, dominate the mesoscale variability in the Caribbean Sea, we will focus on these 57 energetic anticyclones in the remainder of this study.

## 4 Westward intensification

Figure 6 showed that the EKE of the anticyclones with long tracks increases towards the west. We hypothesized that the growth of the anticyclones was governed by a westward strengthening of the horizontal density gradients. In this section, we evaluate the evolution of the horizontal density gradients of the anticyclones and the validity of the thermal wind balance.

Figure 7 shows the different components of the thermal wind balance $(\frac{\partial v}{\partial z}, \frac{g}{\rho_0 f} \frac{\partial \sigma}{\partial x})$ computed from high-pass filtered density and velocity fields. As in Fig. 6c, only the anticyclones with long tracks are taken into account. The vertical shear of the anticyclones (Fig. 7a) increases from the eastern part of the basin towards the west. At $64^o$W and $71^o$W, the vertical shear of the anticyclones increases zonally more rapidly. A comparison to the average shear of the total velocity field indicates that these longitudes are located close to two regions with strong background vertical shear (black contour in Fig. 7a). These two regions coincide with the Cariaco and Guajira upwelling regions, where there is a strong density contrast between the upwelled and surface waters (black contour in Fig. 7b).

The horizontal density gradients of the anticyclones have a similar magnitude and spatial distribution as the vertical shear (Fig. 7b). This indicates that, on average, the anticyclones were close to geostrophy. These horizontal density gradients are due to horizontal temperature gradients (Fig. 7c) and horizontal salinity gradients (Fig. 7d). In this case, the horizontal density gradients due to salinity are slightly bigger than the horizontal density gradients due to temperature.

To gain insight into the westward intensification of the anticyclones, we computed the zonal variation of the EKE, vertical shear, density gradients and properties of the anticyclones (Fig. 8). Overall, the meridional maximum of EKE that is contained in the anticyclones increases from approximately 200 cm$^2$ s$^{-2}$ at $65^o$W towards 530 cm$^2$ s$^{-2}$ at $75^o$W (Fig. 8a). The average westward increase of EKE is larger west than east of $71.5^o$W. A similar difference in westward growth is present in the westward increase of vertical shear and horizontal density gradients (Fig. 8b). In this case, the magnitude of the shear and density gradients increases rapidly between $71.5^o$W and $72.5^o$W. Because the horizontal density gradients and vertical shear are strongly correlated, it confirms that the westward increase is mainly due to geostrophic changes in the flow (Fig. 8b). Although both properties have a similar magnitude, there is a small offset of approximately $0.1 \times 10^{-3}$ s$^{-1}$. This difference indicates that the vertical shear is not fully balanced by the horizontal density gradients, and that there are ageostrophic effects that contribute to the vertical shear of the anticyclones.

As in Fig. 7c and Fig. 7d, the horizontal density gradients due to temperature and salinity both increase towards the west. This increase in both temperature and salinity can be explained by the properties of the upwelled waters, which are both colder and more saline than the background environment. Note that the density gradients induced by salinity are stronger than those induced by temperature differences. Furthermore, the anticyclones become slightly denser on their path westward through mixing with the surrounding waters (not shown). However, this density increase of the anticyclones is weaker than

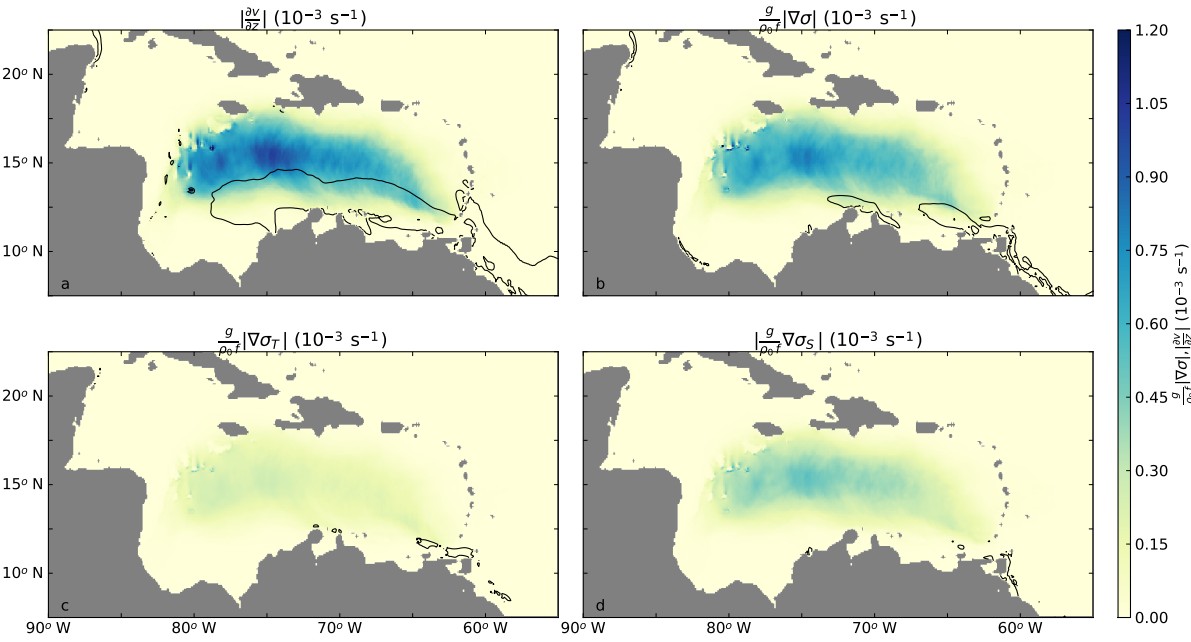

**Figure 7.** (a) Vertical shear of the anticyclones, averaged over the upper 50m. Black contour lines indicate where the strongest background gradients ($8 \times 10^{-3}$ s$^{-1}$) are located. (b) Horizontal density gradients of the anticyclones, scaled with $\frac{g}{\rho_0 f}$ according to the thermal wind equation. (c) Horizontal density gradients due to temperature gradients of the anticyclones. (d) Horizontal density gradients due to salinity gradients of the anticyclones.

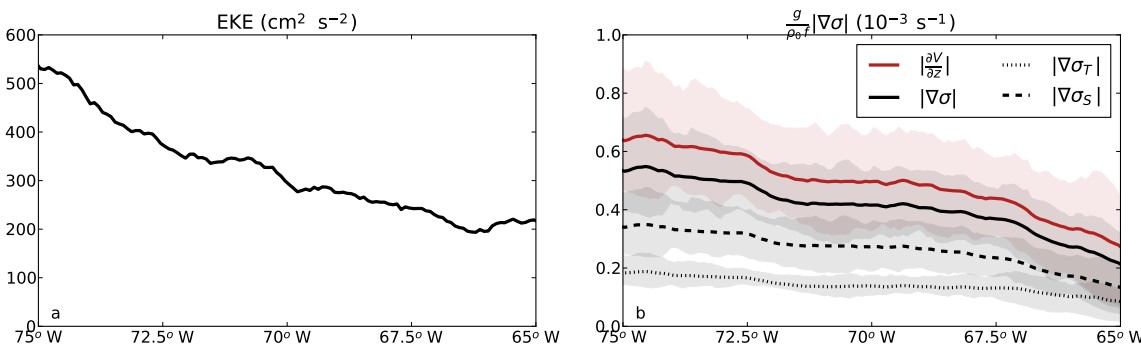

**Figure 8.** (a) Zonal variation of the meridional maximum eddy kinetic energy by the anticyclones with long tracks between 12.5$^o$N and 17.5$^o$N (Fig. 6c). (b) Zonal variation of the vertical shear and horizontal density gradients of the anticyclones with long tracks averaged between 12.5$^o$N and 17.5$^o$N. Shaded regions indicate 25th and 75th percentile. All properties are averaged over the upper 50 m of the water column.





the westward density increase of the surrounding waters, suggesting that indeed changes in properties of the background environment result in the westward increase of the horizontal density gradients.

## 5   Offshore advection of dense upwelling filaments

Figure 8 showed that the westward increase of EKE coincides with a westward increase of the vertical shear and density
gradients of the anticyclones. This indicates that the westward intensification of Caribbean anticyclones follows the thermal wind balance. This westward density increase of the surrounding surface waters is possibly governed by the advection of cold upwelling filaments offshore (Jouanno and Sheinbaum, 2013). According to this study, these filaments, characterized by strong temperature gradients, are advected by the anticyclones. Since we found a sudden increase in horizontal density gradients of the anticyclones at $72^{o}$W in Fig. 8b, we evaluate in this section whether that increase is related to the advection of cold upwelling
filaments.

To study if there is a relation between the filaments and the eddy kinetic energy of the anticyclones, we defined a measure for the strength of the filaments. This measure is a combination of the asymmetry of the anticyclone and the density gradient (induced by temperature) on the western side of the anticyclone, because the filaments are expected on the western side of the anticyclones. The asymmetry is defined as the ratio between the strength of the horizontal density gradient on the western and
eastern side of the anticyclone ($\frac{|\nabla \sigma_{west}|}{|\nabla \sigma_{east}|}$). An asymmetry larger than one corresponds to a situation where the density gradients on the western side of the anticyclone are stronger than on the eastern side, which suggests that an anticyclone advects a filament. Furthermore, we expect that the anticyclones that advect filaments have strong horizontal temperature gradients on the western side. These two properties are shown in Figure 9a. It displays a positive correlation between the asymmetry and the strength of the temperature gradients, so that the most asymmetric anticyclones have the strongest temperature gradients
on their western side, corresponding to our hypothesis that the anticyclones with strong temperature gradients advect cold upwelling filaments.

The relation between the eddy kinetic energy and the advection of filaments is also shown in Figure 9a. The average eddy kinetic energy is computed on the eastern side of the anticyclone to exclude the kinetic energy associated with the filament. The EKE of the anticyclone is positively correlated to the strength of the western temperature gradient of the anticyclone, which
denotes that the anticyclones with strong filaments are more energetic than the anticyclones with weaker filaments. This means that the westward intensification of individual anticyclones can be affected by the advection of cold filaments.

To understand this positive correlation between the advection of filaments and the strength of the anticyclones, we compared the anticyclones with long tracks to the other anticyclones that are present in the Caribbean Sea between $65^{o}$W and $75^{o}$W (black and grey lines in Fig. 6, respectively). This comparison between the asymmetry of the anticyclones, their life time and
their meridional location reveals two interesting aspects (Fig. 9b). First, the anticyclones with long tracks (dots in Fig. 9b) are located in the southern part of the basin ($<16^{o}$N). Second, the anticyclones with long tracks are in general more asymmetric than the other anticyclones (triangles in Fig. 9b).




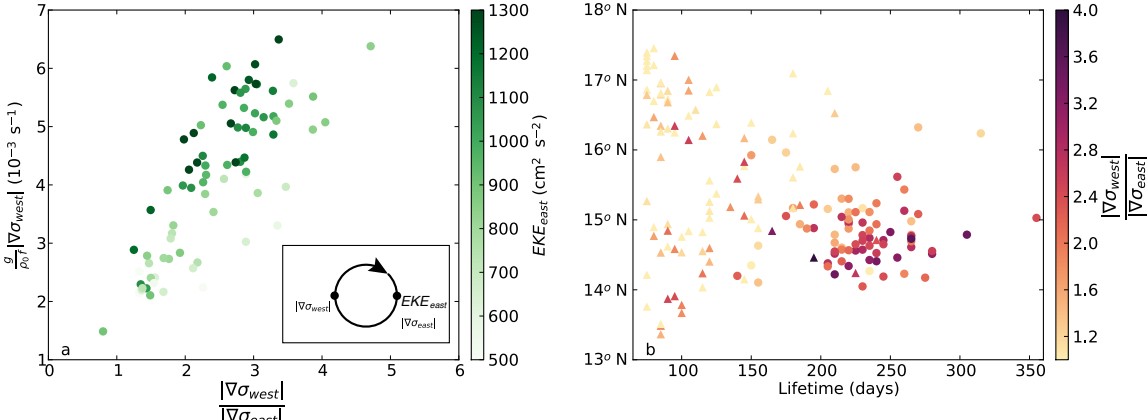

**Figure 9.** (a) Asymmetry of the anticyclones ($\frac{|\nabla\sigma_{west}|}{|\nabla\sigma_{east}|}$) versus the strength of the horizontal temperature gradients on the western side of the anticyclones ($|\nabla\sigma_{T,west}|$) in Ekman100. The asymmetry is defined as the ratio between the strength of the horizontal density gradient on the western and eastern side of the anticyclone between $65^{o}$W and $75^{o}$W. Colors indicate the EKE of the eastern side of each eddy. Each dot represents one anticyclone, the inlay contains a schematic of the locations where the properties are calculated. (b) The total life time versus latitude of each anticyclone. Anticyclones with long tracks (black lines in Fig. 6a) are indicated with dots, anticyclones with short tracks (grey lines in Fig. 6a) are indicated with triangles. Colors denote the asymmetry of the anticyclones. All properties are calculated as average values between $65^{o}$W and $75^{o}$W from individual anticyclones in all simulations.

The combination of these two aspects suggests a relation between the advection of filaments and the lifetime of Caribbean anticyclones: Anticyclones that propagate close to the upwelling region are more likely to advect a cold filament that increases its western horizontal temperature gradients. Such offshore advection of dense filaments increases the horizontal density gradients leading to the westward intensification of the anticyclones. Consequently, these anticyclones become more energetic and

5   their lifetime increases. In contrast to the anticyclones with long tracks, the anticyclones with short tracks (triangles in Fig. 9b) are in general more symmetric and located farther north, and thus apparently do not advect cold filaments. This suggests that anticyclones that do not advect cold filaments dissipate, deform or merge rather than intensify (van der Boog et al., 2019). This is in line with the earlier result that the anticyclones with long tracks are more energetic than the anticyclones with short tracks (Fig. 6c,d).

10  **6  Impacts of varying upwelling**

In the previous section, we found that the westward growth of Caribbean anticyclones follows the thermal wind balance, and is characterized by an increase of horizontal density gradients of the anticyclones, due to the advection of cold upwelling filaments by the anticyclones themselves. Therefore, we expect that changes in upwelling strength result in changes in the properties of the filaments, which would affect the westward growth of these anticyclones. To study the impacts of upwelling



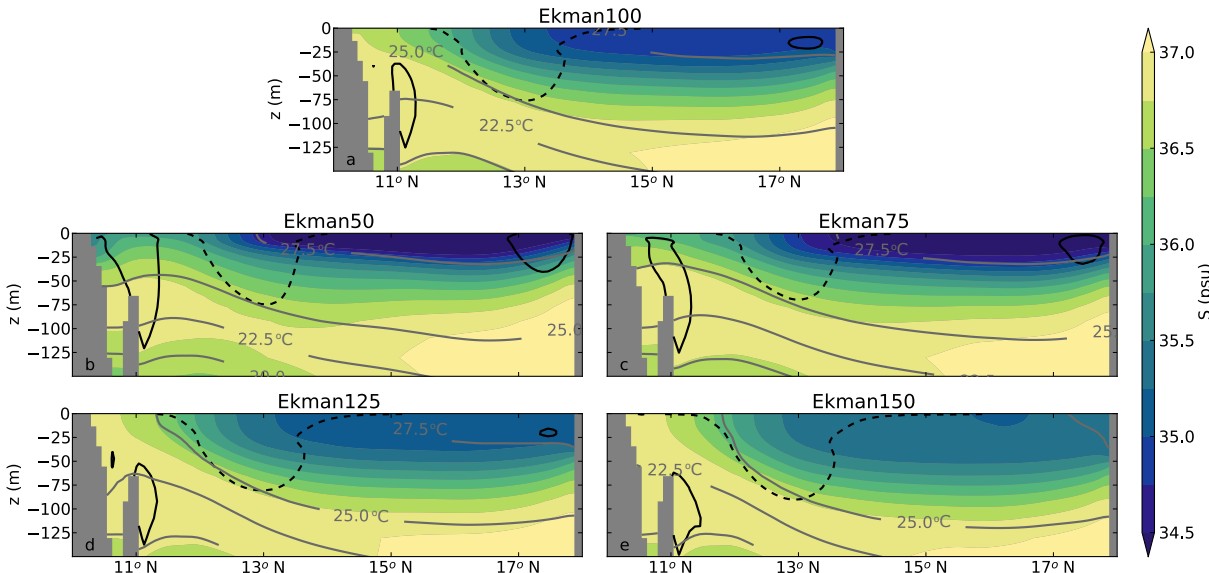

**Figure 10.** Cross-section at $66^o$W of the 20-year averaged salinity in (a) Ekman100, (b) Ekman50, (c) Ekman75, (d) Ekman125 and (e) Ekman150. The solid black line indicates the position of the subsurface countercurrent with a westward velocity of 0.07 m s$^{-1}$. The Caribbean Current is indicated with the dashed solid line that shows a westward velocity of 0.2 m s$^{-1}$. The grey contour shows the 20-year average temperature in each simulation.

on this westward growth of the anticyclones, we performed sensitivity simulations in which we altered the upwelling strength (Section 2). This resulted in differences in both the mean flow and mesoscale variability.

## 6.1 Changes in upwelling

The zonal wind stress was decreased with a constant factor in Ekman50 and Ekman75 corresponding to a theoretical northward
5    Ekman transport of 50% and 75% compared to Ekman100, respectively. The decrease in zonal wind stress in Ekman50 and Ekman75 results in weaker upwelling and consequently warmer SST in Cariaco Basin (Fig. 10b,c). The highest coastal temperatures are present in Ekman50. Sea-surface salinity decreases in both Ekman75 and Ekman50 compared to Ekman100 (Fig. 10b,c): this freshening is related to the presence of a subsurface salinity maximum in the Caribbean Sea, causing upwelled waters to be more saline than surface waters. Weaker upwelling thus results in warmer and fresher surface waters (Fig. 10b,c),
10   and stronger upwelling results in colder and more saline surface waters (Fig. 10d,e).

Besides differences in the mean properties ($\sigma, T, S$), the width of the upwelling region also changes in the simulations: In Ekman125 and Ekman150, the upwelling is confined to a narrower region than in Ekman75 and Ekman50. Moreover, we find that the core of the subsurface countercurrent is displaced upwards in Ekman50 and Ekman75 compared to Ekman100 (black contour in Fig. 10). This upward change in line with the observations where the core was also displaced upwards in weaker





wind conditions (Andrade, 2003; Andrade and Barton, 2005). The increased northward Ekman transport in Ekman125 and Ekman150 leads to colder and more saline surface waters. In these simulations, the core of the subsurface countercurrent is positioned lower in the water column (black contour in Fig. 10). The changing position of the subsurface countercurrent and the changes in sea-surface properties in the upwelling regions indicate that the model is able to reproduce realistic upwelling

properties as was also shown in Section 3.

## 6.2   Changes in EKE

The upwelling strength impacts the offshore EKE (Fig. 11). In the interior of the Caribbean Sea, the surface EKE and subsurface EKE decreases (increases) for weaker (stronger) zonal wind strength. For example, the magnitude of the mean EKE in the interior of the Caribbean Sea decreases by 27% in Ekman50 compared to Ekman100 (Fig. 11a). In Ekman75, a slightly weaker

decrease of 15% is present (Fig. 11b). In Ekman125 and Ekman150, the surface EKE increases with 13% and 28%, respectively (Fig. 11c,d). Overall, the EKE variations indicate a positive correlation between the coastal wind stress and the strength of the mesoscale eddies. A similar correlation is found in the subsurface EKE. This is in line with the finding that the upwelling was shallower in the simulations with weaker upwelling and deeper in simulations with stronger upwelling.

To gain insight into the westward growth of EKE, the meridional maximum of EKE from all mesoscale variability between

$12.5^o$N and $17.5^o$N was computed as a function of longitude (Fig. 12a). The largest changes in EKE are seen in the western part of the basin. In Ekman100, the EKE increases with 123% from $65^o$W to $75^o$W, while the weaker upwelling in Ekman50 and Ekman75 result in a smaller westward increase of the EKE, 62% and 97%, respectively. Ekman125 and Ekman150 have a stronger westward increase in EKE compared to Ekman100. Overall, there is a clear positive correlation between the upwelling strength and the westward increase in EKE.

## 6.3   Changes in eddy characteristics

Next, we assess whether the changes in the westward increase in EKE can be attributed to changes in the eddy field and to changes in the westward intensification of anticyclones. To this end, we studied the behavior of individual eddies in the each simulations with the py-eddy tracker. The number of anticyclones that are formed each year in the interior of the Caribbean Sea varies between 8.55 year$^{-1}$ (Ekman100) and 10.85 year$^{-1}$ (Ekman50, Table 1). These numbers are in line with a formation

rate of 8-12 anticyclones per year as observed by Richardson (2005). Of the anticyclones, 34-43% are formed in the eastern part of the basin, and are tracked further west of $75^o$W (Table 1). The paths of these anticyclones with long tracks are, similar to Ekman100, located in the southern part of the basin (not shown).

Similar to Ekman100, we computed the EKE associated only with the anticyclones that are formed in the eastern part of the basin ($<65^o$W) and propagate beyond $75^o$W (Fig. 12b). Similar behavior is visible for the total EKE in Figure 12a: The

EKE increases towards the west in all simulations, and stronger (weaker) upwelling correspond to a larger (smaller) westward increase. In the simulations with stronger upwelling (Ekman100, Ekman125 and Ekman150), the anticyclones with long tracks (Fig. 12b) are responsible for more than half of the total EKE (Fig. 12a), even though they are only due to 30-40% of the total number of anticyclones in this region (Table 1). In Ekman50 and Ekman75, the anticyclones with long tracks become less



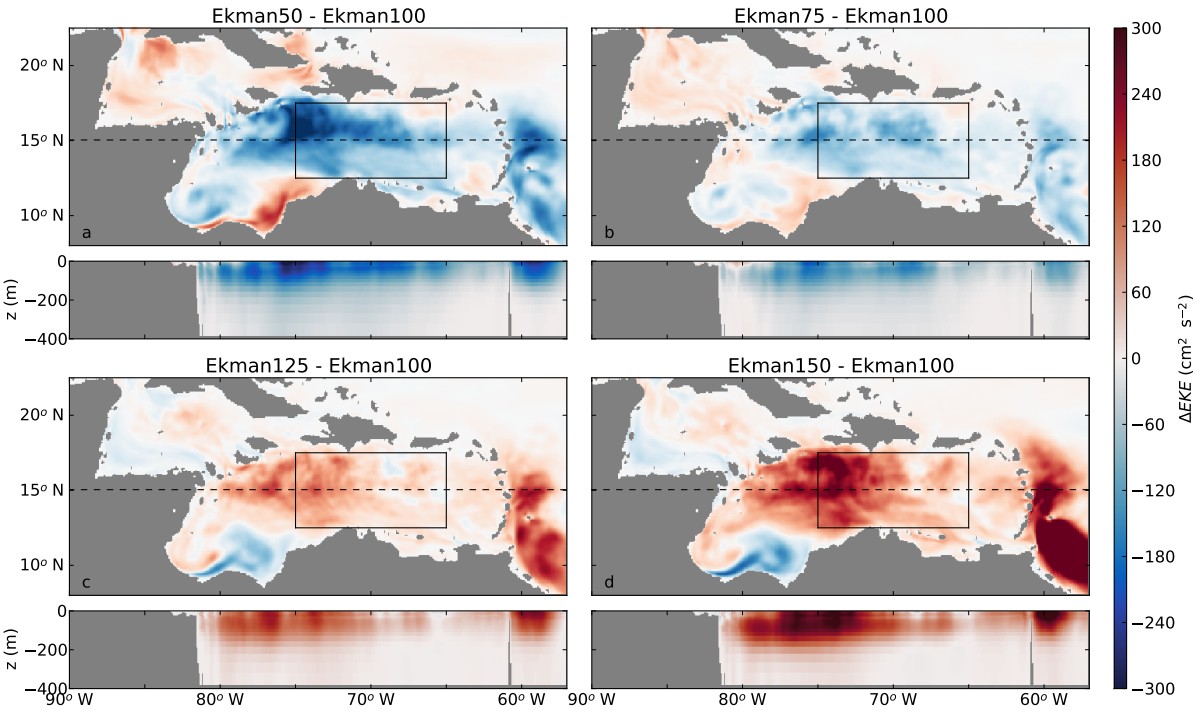

**Figure 11.** EKE anomaly compared to Ekman100 (Fig. 4) at the near-surface and at a cross-section at $15^oN$ in (a) Ekman50, (b) Ekman75, (c) Ekman125 and (d) Ekman150. Positive values correspond to enhanced EKE compared to Ekman100. The black box indicates the interior of the Caribbean Sea ($12.5^oN$-$17.5^oN$, $65^oW$-$75^oW$). The dashed line shows the location of the cross-sections in the lower panels.

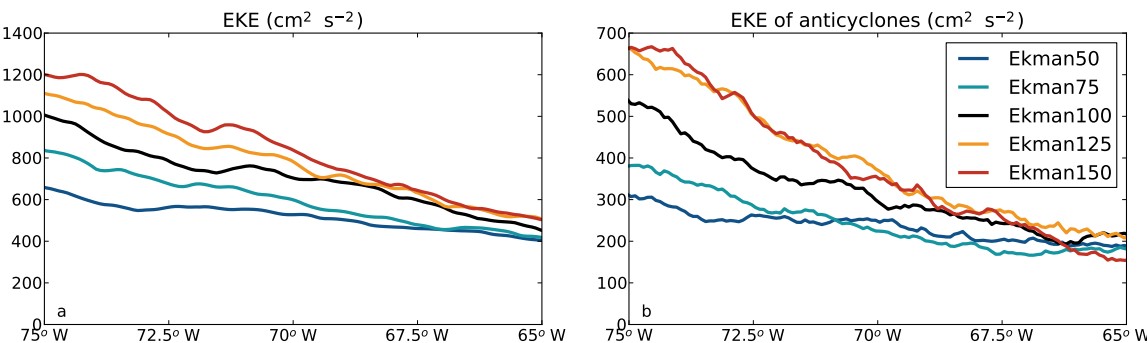

**Figure 12.** (a) Zonal variation of the maximum of EKE between $12.5^oN$-$17.5^oN$, from $65^oW$ to $75^oW$ for each simulation. (b) Zonal variation of the meridional maximum of EKE between $12.5^oN$-$17.5^oN$ of the anticyclones with long tracks. The EKE was averaged over the upper 50 m. Note the difference in scale between the two panels

.





dominant. In these simulations, the westward increase of EKE is also less pronounced than in the stronger wind conditions. Overall, this shows that in all simulations, a substantial part of the EKE and the longitudinal variations are governed by the evolution of a small number of anticyclones and that this effect is stronger in simulations with stronger upwelling.

In general, the anticyclones are fresher and warmer than the surrounding waters in all simulations. The freshest and warmest anticyclones are present in Ekman50, while the most saline and coldest anticyclones are found in Ekman150 (Table 1). It is interesting to note that the variation of the properties of the anticyclones is very small in each simulation (Table 1). This suggests that the properties of the anticyclones remain approximately constant during the westward propagation.

Similar to observations (Centurioni and Niiler, 2003), we found less cyclones than anticyclones in each simulation (Table 1). The lowest formation rate of cyclones is found in Ekman150 with 4.15 cyclones per year, and the highest formation rate is present in Ekman50 with 6.15 cyclones per year. In none of the simulations, the py-eddy tracker was able to track multiple cyclones from east to west ($65^o$W-$75^o$W). This implies that the cyclones are either deformed or dissipated too much, such that the py-eddy tracker could not track their sea-level anomaly. Overall, the behavior of the mesoscale eddies is similar in all simulations and the spatial pattern and magnitude of the surface EKE is governed by the anticyclones with long tracks. The westward intensification of these anticyclones is discussed in the next section.

## 6.4 Westward intensification

In Section 4, we showed that the westward increase of EKE in Ekman100 coincided with a westward strengthening of the vertical shear of the most energetic anticyclones (Fig. 6c, Fig.7a). This increase in vertical shear was related via the thermal wind relation to the strengthening of the horizontal density gradients at the western edge of the anticyclone caused by the advection of upwelling filaments (Fig. 9a). Here, we analyse the thermal wind relation in the other simulations and its relation to the westward growth of the anticyclones (Fig. 13).

In all simulations, the vertical shear of the anticyclones increases from east to west (Fig 13a). This westward increase in vertical shear is similar to the westward strengthening of the horizontal density gradients (Fig. 13b). Similar to Ekman100, the magnitude of the horizontal density gradients is slightly smaller than the vertical shear in all simulations, but their variation is similar. In combination with the low Rossby number of the anticyclones (Table 1), this implies that the anticyclones are in near-geostrophic balance during their evolution.

The anticyclones display the strongest westward growth in the simulations with stronger upwelling (orange and red lines in Fig. 13a,b). These simulations have relatively low vertical shear at $65^o$W. In the simulations with weaker upwelling, Ekman50 and Ekman75, the anticyclones intensify less than in Ekman100. Moreover, in Ekman50, the vertical shear even weakens between $70^o$W and $73^o$W. At the same location, the EKE also did not increase in Ekman50 (Fig. 12b).

It is remarkable that, at $65^o$W, the strongest vertical shear and density gradients of the anticyclones are present in Ekman50, while the weakest gradients were located in Ekman150 (compare the blue and red lines in Fig. 13a,b). This suggests that the early development (east of $65^o$W) of the anticyclones differs between the simulations. The strong gradients in Ekman50 at $65^o$W are surprising, because we expected stronger gradients in simulations with stronger upwelling. An explanation for the strong horizontal density gradients at $65^oW$ in Ekman50 can be found after separating the total horizontal density gradients





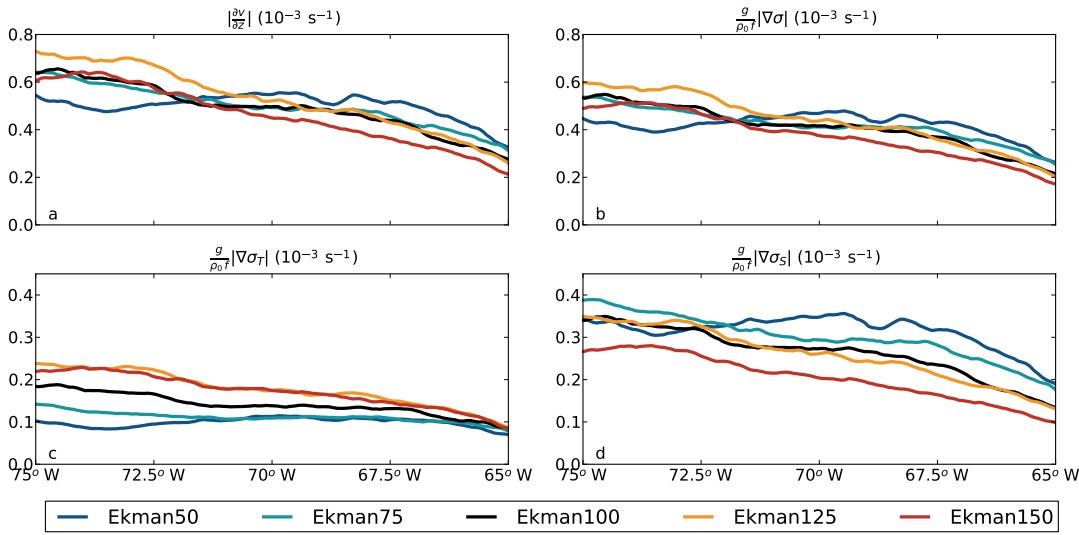

**Figure 13.** Zonal variation of the meridional average ($12.5^oN$-$17.5^oN$) of the (a) average vertical shear of the anticyclones, (b) horizontal density gradients, (c) horizontal density gradients induced by temperature gradients and (d) horizontal density gradients induced by salinity gradients. All properties are computed over the upper 50 m of the water column.

into density gradients induced by temperature and salinity (Fig. 13c,d). While the density gradients driven by temperature gradients have a similar magnitude in each simulation in the eastern part of the basin (Fig. 13c), the density gradients driven by salinity gradients differ substantially between the simulations at $65^oW$ (Fig. 13d) and are negatively correlated to the upwelling strength.

To understand why the salinity gradients in the eastern part of the basin are stronger for weaker upwelling, we analysed the average spatial variation of the total density gradients. Figure 14 shows the magnitude of the time-averaged horizontal density gradients $|\nabla\sigma|$ of Ekman100 (middle column) and the strengthening or weakening of these gradients in Ekman50 (right column) and Ekman150 (left column) compared to Ekman100. The strongest horizontal density gradients are located at the Guajira and Cariaco upwelling regions (Fig. 14b). These density gradients are weaker in Ekman50 than in Ekman100 (Fig.

14a), and stronger in Ekman150 (Fig. 14c).

The horizontal density gradients at the upwelling region are due to both horizontal temperature and salinity gradients (Fig. 14e-h). Because upwelling brings relatively cold and saline waters towards the surface, the strength of these density gradients increase for stronger upwelling (Fig. 14c) and decrease for weaker upwelling (Fig. 14a). In Ekman150, the strengthening of the horizontal density gradients is mainly due to temperature induced density gradients in the upwelling region (Fig. 14f,i).

In contrast, in Ekman50, the weakening of the temperature gradients (Fig. 14d) and salinity gradients (Fig. 14g) contribute roughly equally to the weakening of the total horizontal density gradients (Fig. 14a).

In the interior of the basin, the horizontal density gradients strengthen compared to Ekman100 in Ekman50 (Fig. 14a), while they are weaker in Ekman150 (Fig. 14c). This response results from the increasing importance of the Amazon and Orinoco


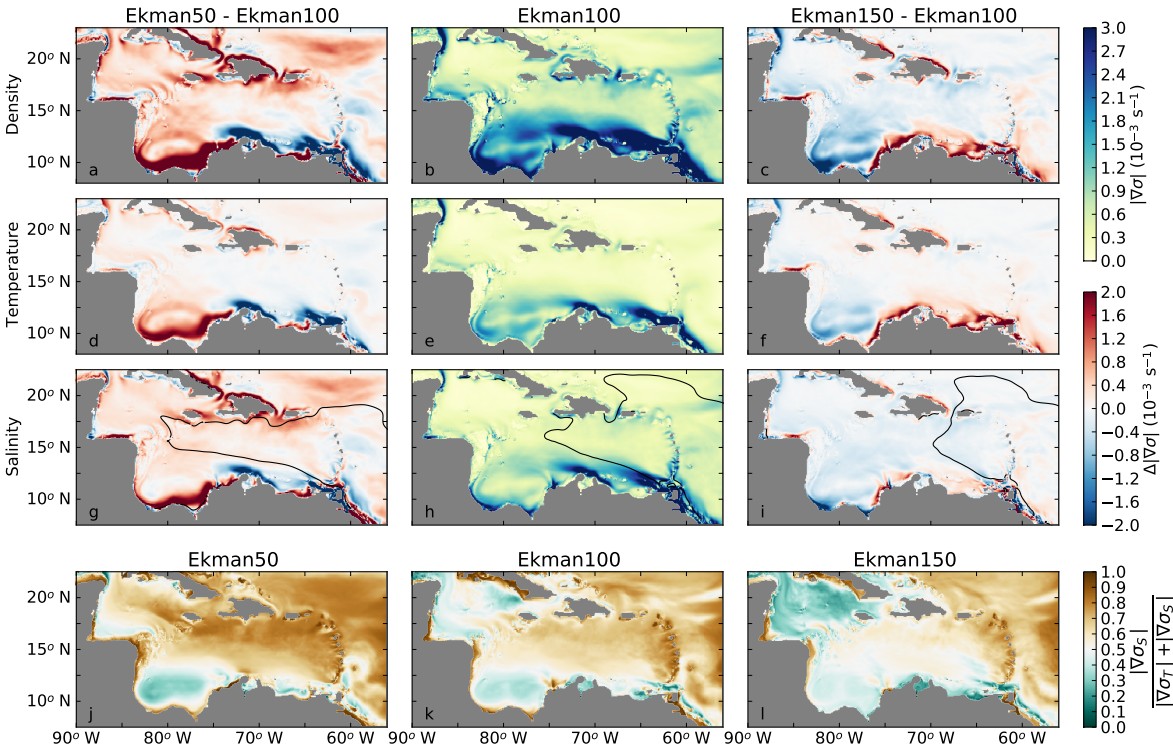

**Figure 14.** (a-c) Average strength of the horizontal density gradients ($|\nabla\sigma|$) in Ekman50, Ekman100, Ekman150 (see Section 2). The middle column shows the magnitude of the density gradients in Ekman100. Left and right columns contain the difference between Ekman50 and Ekman150 and Ekman100, respectively. Red colors indicate an increase of the horizontal density gradient. (d-f) Average strength of the horizontal density gradients due to temperature variations ($|\nabla\sigma_T|$). (g-i) Average strength of horizontal density gradients due to salinity variations ($|\nabla\sigma_S|$). The solid black line indicates where $S = 35.5$ psu. (j-l) Relative magnitude of density gradients due to salinity compared to density gradients due to temperature ($\frac{|\nabla\sigma_S|}{|\nabla\sigma_T|+|\nabla\sigma_S|}$). All properties are averaged over 20-years and over the upper 50 m of the water column.





River plumes for weaker wind conditions. The wind affects both the propagation and spreading of the river plume. First of all, the northward Ekman transport advects the river plumes towards the north as previously observed by Molleri et al. (2010). This northward Ekman transport is weaker in Ekman50, and consequently the river plumes follows the mean flow towards the west (Fig.14g). The stronger Ekman transport in Ekman150 results in a more northward path of the river plumes (Fig.14i). The

second consequence of changing wind conditions is the impact of the wind-driven vertical mixing. The strong zonal winds in Ekman150 induce more wind-driven mixing of the surface waters than the winds in Ekman50. As a result, salinity anomalies propagate less far westward in Ekman150 than in Ekman50. The farther advection of the river plume in Ekman50 explains why the anticyclones have stronger salinity gradients in Ekman50 than in the other simulations (Fig. 13d).

From this analysis we conclude that a weakening of the zonal wind in Ekman50 leads to an increased influence of the

Amazon and Orinoco River plume compared to the other simulations. More specifically, in Ekman50, the upper ocean density gradients in the interior of the Caribbean Sea are dominated by salinity gradients (Fig. 14j), while the density gradients are predominantly temperature driven in Ekman150 (Fig. 14l). So, depending on the zonal wind strength, the total horizontal density gradients in the Caribbean Sea are either driven by temperature or salinity gradients, which suggests that both the river plumes as well and the upwelling affect the variability in the basin.

## 15   7   Summary and discussion

In this study, we used a regional model of the Caribbean Sea to investigate the interaction between mesoscale anticyclones and the wind-driven coastal upwelling along the South-American coast. We showed that the westward intensification of Caribbean anticyclones is driven by an increase in the horizontal density gradient between their cores and their surroundings (Fig. 7). Notably, the increase is governed by density changes of the surroundings; not by density changes within the anticyclone.

More specifically, the density of the surroundings increases through the offshore advection of cold filaments of upwelled water by the anticyclones, which in turn is governed by the passage of the anticyclones themselves. This increases the horizontal density gradient between the anticyclone and the surroundings on the western side of the anticyclone (Fig. 9a). As a result, its vertical shear increases, and the anticyclone becomes more energetic. Approximately 2-4 anticyclones per year showed this behavior. The anticyclones that are not associated with the advection of cold filaments, strengthen this view that the westward

intensification of Caribbean anticyclones is facilitated by the advection of filaments of upwelled water. These anticyclones were less energetic and had shorter life spans than the anticyclones that intensified towards the west.

Further support for this view is obtained from the series of simulations in which the strength of the upwelling was altered by adjusting the zonal wind stress (and thus the northward Ekman transport). Stronger upwelling leads to an increase in offshore cooling, and thus to a stronger increase in the horizontal density gradient between the anticyclones and their surroundings. As a

result, the anticyclones intensify more on their path westward compared to the case of weaker upwelling. We also found that a decrease of the northward Ekman transport in the basin allows the river plumes to advect further into the basin. This influences the early development of the anticyclones. This non-linear response of the evolution of anticyclones to changes in upwelling





highlights the fact that the horizontal density gradients in the Caribbean Sea are predominantly set by salinity gradients in case of weak wind forcing and by temperature gradients in case of strong wind forcing.

Previous studies in the Caribbean Sea mainly focussed on either the influence of the river plumes (e.g., Hu et al., 2004; Chérubin and Richardson, 2007), or on the influence of the wind (e.g., Oey et al., 2003; Astor et al., 2003; Jouanno et al.,

2009). However, based on our results, we argue that it is important to take the interaction between the pathway of the river plume and the upwelling into account when studying mesoscale variability in the Caribbean Sea. Furthermore, we showed how the westward intensification of Caribbean anticyclones could be driven by baroclinic instabilities, as suggested by (Carton and Chao, 1999; Jouanno et al., 2009).

Based on the different wind forcing in the simulations, we can speculate on the seasonal eddy variability in the Caribbean

Sea: the strong zonal wind stress in winter is similar to Ekman125, and the weaker winds in fall are represented by Ekman50 and Ekman75. In Ekman150, the anticyclones intensified more than in Ekman50. This implies that during the strong wind forcing in winter the anticyclones intensify more. Furthermore, during weak wind forcing, we expect that the salinity gradients of the river plume will become dominant. However, the dispersal of the Amazon and Orinoco River plumes also has a distinct seasonal cycle (Hellweger and Gordon, 2002; Chérubin and Richardson, 2007), which is absent in our model.

The results of this study also highlight some aspects of the interannual variability of the eddy field in the Caribbean Sea. On interannual time scales, our results suggest a possible positive correlation between the wind forcing and eddy variability in the interior of the basin. Jouanno and Sheinbaum (2013) used a model with seasonally varying boundary conditions and identified a similar relationship. This is also found in observations (Fig. 15), which show that sea-surface variance is higher in years with stronger zonal winds. Figure 15 also suggests that the response of the sea-surface variance to the wind stress is non linear:

Although 2010 and 2011 were years with weak zonal winds, the sea-surface variance was relatively high. It is interesting to note that salinity was observed in the Cariaco Basin was anomalously low in these years (Cariaco project , 2019). Taking into account the shallow depth of this salinity anomaly, it is plausible that it is related to the farther westward propagation of the river plumes as seen in Ekman50.

Over the past two decades, a weak decreasing trend has been observed in the coastal wind stress and upwelling strength

(Campbell et al., 2011; Lima and Wethey, 2012; Torres and Tsimplis, 2013). Not only did this have severe ecological impacts (Villamizar G. and Cervigón, 2017), it also impacts the mesoscale variability. Weaker upwelling will lead to less westward intensification of Caribbean anticyclones and less mesoscale variability in the interior of the basin. Furthermore, in weaker wind conditions the offshore advection of (nutrient-rich) cold filaments is weaker, which results in less offshore advection of nutrients and biota.

Overall, in this study we showed how large-scale processes in the Caribbean Sea impact the eddy variability in this basin. We showed how two processes induced by the northward Ekman transport impact the development of the most energetic anticyclones. First, the river plume sets the properties of the anticyclones during their early development in the eastern Caribbean Sea. Second, the anticyclones intensify themselves by the advection of cold and saline upwelling filaments. Together these two processes explain the mesoscale variability in the Caribbean Sea.





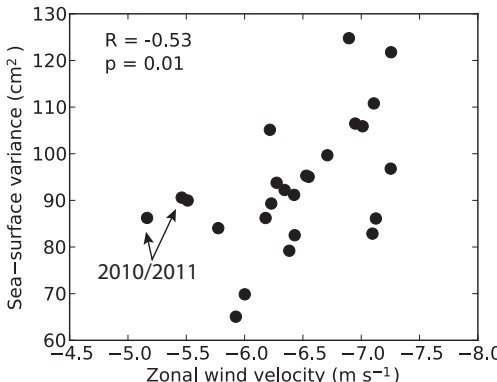

**Figure 15.** Year-averaged sea-level variance and coastal zonal wind velocities. Sea-surface spatial variance is obtained from daily sea-level anomaly fields from years 1993-2017 from satellite altimetry (CMEMS) and averaged over the interior of the Caribbean Sea (12.5°N - 17.5°N, 65°W - 75°W). Coastal zonal winds in the upwelling region (10°N -12.5°N, 65°W - 75°W) are obtained from years 1993-2017 from ECMWF (ERA-Interim, Dee et al., 2011). Each dot represents one year.

*Code and data availability.* For information about the regional configuration of MITgcm and about the processing code of all simulations, contact Carine van der Boog. The boundary conditions for the regional setting were obtained from the Operational Mercator global ocean analysis product provided by the Copernicus Marine Environment Monitoring System

*Author contributions.* Carine van der Boog designed the model and computational framework and analyzed the data. Caroline Katsman,
5 Julie Pietrzak and Henk Dijkstra supervised the work and helped shape the research, analysis and manuscript. Nils Brüggemann provided valuable support on the configuration of the model. All other authors discussed the results and contributed to the final manuscript.

*Competing interests.* The authors declare that they have no conflict of interest.

*Acknowledgements.* The work of Carine van der Boog is financed by a Delft Technology Fellowship awarded to Caroline Katsman. This work is part of the research program ALW-Caribbean with project 858.14.061 (SCENES), which is financed by the Netherlands Organisation
10 for Scientific Research (NWO). We would like to thank Sabine Rijnsburger for her suggestions to analyse the effect of the river plume. We would also like to thank Sotiria Georgiou, Adam Candy, Steffie Ypma and Juan Manual Sayol for valuable discussions on this work.





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





**Table 1.** Number of eddies per year and average Rossby number with standard deviations in each simulation between $12.5^oN$-$17.5^oN$, $65^oW$-$75^oW$. Amounts are obtained with the py-eddy tracker. The formation rate is computed as the number of eddies that are formed each year east of $75^oW$. The long tracks represents the percentage of the eddies that were formed east of $65^oW$ and could be tracked west of $75^oW$. The Rossby number is calculated as $Ro = u_{swirl}/(R_{eddy} * f)$, where $u_{swirl}$ is the swirl velocity of the eddy, $R_{eddy}$ is the radius and $f$ the Coriolis parameter. Density, temperature and salinity is taken at the core of the anticyclones. One standard deviation is added to indicate the interannual variability.

|  | Ekman50 | Ekman75 | Ekman100 | Ekman125 | Ekman150 |
|---|---|---|---|---|---|
| **Anticyclones** |  |  |  |  |  |
| Formation rate (year$^{-1}$) | 10.85±2.22 | 9.25±1.76 | 8.55±1.72 | 8.65±1.19 | 9.10± 1.89 |
| Long tracks ($<65^oW$, $>75^oW$) | 36% (71/199) | 34% (57/167) | 37% (57/153) | 43% (70/160) | 36% (59/165) |
| Rossby number | 0.14±0.01 | 0.14±0.01 | 0.15±0.01 | 0.15±0.02 | 0.15±0.01 |
| Density (kg m$^{-3}$) | 21.94±0.11 | 22.20±0.12 | 22.68±0.09 | 22.90±0.05 | 23.05±0.06 |
| Temperature ($^oC$) | 27.93±0.02 | 27.79±0.03 | 27.64±0.05 | 27.52±0.05 | 27.39±0.06 |
| Salinity (psu) | 34.09±0.14 | 34.38±0.14 | 34.96±0.11 | 35.19±0.05 | 35.34±0.06 |
| **Cyclones** |  |  |  |  |  |
| Formation rate (year$^{-1}$) | 6.15 ± 1.49 | 5.30 ± 1.62 | 5.70 ± 1.71 | 4.90 ± 1.67 | 4.15 ± 2.06 |
| Long tracks ($<65^oW$, $>75^oW$) | 0% (0/105) | 2% (2/89) | 1% (1/95) | 0% (0/79) | 0% (0/69) |
| Rossby number | 0.11 ± 0.02 | 0.12 ± 0.02 | 0.14 ± 0.03 | 0.15 ± 0.04 | 0.12 ± 0.03 |
| Density (kg m$^{-3}$) | 23.17±0.15 | 23.12±0.28 | 23.44±0.14 | 23.51±0.25 | 23.49±0.27 |
| Temperature ($^oC$) | 27.49±0.14 | 27.40±0.24 | 27.09±0.18 | 26.94±0.36 | 26.95±0.40 |
| Salinity (psu) | 35.54±0.16 | 35.44±0.28 | 35.73±0.11 | 35.76±0.18 | 35.74±0.20 |