# Peer review of "The Impact of Upwelling on the Intensification of Anticyclonic Ocean Eddies in the Caribbean Sea"

_Ocean Science, 2019_

## Referee Comment (RC1) · Anonymous Referee #1 · 15 Aug 2019

The comment was uploaded in the form of a supplement:
https://www.ocean-sci-discuss.net/os-2019-51/os-2019-51-RC1-supplement.pdf
* * *

---

## Referee Comment (RC2) · Anonymous Referee #2 · 19 Aug 2019

In this manuscript the authors use a numerical simulation to study the effect of upwelling filament advected by anticyclonic eddies in the eastern Caribbean basin. They showed that upwelling filaments, mostly from the upwelling centers are entrained on the western side of the eddies and contribute to their westward intensification. The intensification gets stronger when the upwelling is stronger and vice and versa but it can be influenced by the low salinity of the Amazon and Orinoco plume, which is also influenced by the wind. I think that this study in a good contribution to the understanding of the dynamics of the circulation in the Caribbean Sea and the authors have made an excellent job at describing the mechanics of the intensification process within the case studies that they analyzed. But it would have been good to show in a simulation

with realistic forcing that reproduce the seasonal cycle, that ACs intensifies as they propagate westward.

Most of my comments are technical and there are a few sentences that I wasn't able to understand that need to be properly edited. And it starts with the abstract.

Abstract edits - Should read "These dense filaments are advected by the 5 anticyclones, leading to an increase of the horizontal density gradients on the western side of the anticyclones". - Should read "Following the thermal wind balance, the increased density gradients result in an increase of the vertical shear of the anticyclones and of their westward intensification." - "As expected, stronger (weaker) upwelling is associated with more stronger (weaker) offshore cooling and a more (less) westward". Remove "more". - Last sentence is difficult to understand. Should be rephrased. Especially without the context of the study. Think of someone who'd just reading the abstract, it would be unable to follow.

Manuscript edits Page 2: " and propagate along the wind-driven upwelling regions along the South-American coast". Remove the "along" repetition.

Page 3, lines 10-20: what topography was used and how the passages between the island were accounted for in the 1/12-degree resolution grid? Was the transport between the island estimated and compared to observations? How did the model perform in the other passages?

Page 3, line 20-25: It is difficult to understand how the forcing were applied. It is said that stationary conditions were applied, but it is not clear exactly which ones. For instance the SST relaxation was not stationary? Was it released to monthly averaged SST? What about heat and freshwater fluxes?

Page 3, last paragraph: Did all simulations used stationary boundary conditions? Not only Ekman 100, right?

Page 3, line 30: I don't understand the sentence: "the upwelling regions corresponds
to of the year-averaged northward Ekman transport (100%),". How is the year averaged calculated? Is it over the 20-year simulation? But which simulation since in your case studies the simulations cannot be realistic because of the stationary boundary conditions?

Page 4, Line 5: "with a constant proportional to the wind stress at the upwelling regions". Not sure the sentence expresses what the authors are trying to say. The authors mean to talk about the wind stress magnitude that was reduced by the same amount between simulations. Please revise.

Page 4, line 14: please indicate the figure that shows the upwelling centers location.

Page 5, first line: "We will use this set of simulations with the different to study aspects of the seasonal and". Please revise.

Page 5: "the swirl velocity as the average of the maximum northward and maximum southward velocity of the eddy". Is the location estimated from the center of the SSH anomaly or the point of maximum SHH? Also, why using the meridional velocity only? Why not the location of the maximum of the radial velocity instead?

Page 6, line 3: the thermal wind balance is not a force or a driver, but rather another way to express the geostrophic balance. So, it is meaningless to say how it affected the westward intensification. It is simply the horizontal density gradient as you express it. It can be related to the vertical shear through the thermal wind balance equations.

Pages 6, line 9: Can you show how sigma\_T and sigma\_S contribute to sigma? What equation of state was used?

Page 6, line 18: should read "Further west, ... at 17N.." maybe the longitude can be given here as well because the sentence starts with "Further west ...".

Page 7, line 11: I think the authors meant Fig. 3c.

Page 7, lines 12-15: over what depth this density gradient can be observed. I would
imagine that it strongly depends on the thickness of the fresh water plume? All the dynamics discussed in this study is limited to the first 50 meters, which is the vertical extension of most eddies. What happens below? Do the surface eddies have a deep signature and are they also intensified at depth?

Page 8, lines 10: "corresponds" could be replaced by "matches".

Figure 5, caption: "Near-surface properties of the Caribbean Sea, averaged over 5 days in Ekman100 in year 20"

Page 9, line 17: Not sure the first sentence is correct here. None of what it says has been proven yet. Page 10, line 6: Do the authors mean "To assess the contribution of the anticyclones with the long tracks to the total EKE variability..."?

Page 11 line: which component of the velocity is used? Only v or the magnitude?

Page 11, line 9: "At 64oW and 71oW, the vertical shear of the anticyclones increases zonally more rapidly. " I have a problem with these statement. It is based on visual assessment, which is difficult to prove. Figure 8 could show that, but it only starts at 65W, so one can't see the strong increase. Also, the strong increase at 71W is not visible. Then it becomes harder to see the link with the upwelling centers, although I think the link between the sharp shear increase and the upwelling filament average position is a viable argument. Maybe Figure 8 can be expanded to show that?

Page 11, line 10: "A comparison to the average shear of the total velocity field indicates that these longitudes are located close to two regions with strong background vertical shear (black contour in Fig. 7a)." I don't understand what is done here.

Page 11, lines 15-16: what else than temperature and salinity the density gradient could be due to?

Page 11, line 21: to make such statement, which is not obvious in Figure 8, the slopes along the curve could be shown on Figure 8

OSD
Page 11, line 26-27: it means that AC are not fully geostrophic.

Page 11, line 31: cite Figure 7(c&d) after "...differences."

Figure 8: the average AC density anomaly could also be shown and the figure should start at 64W.

Page 13, line 9: why is the location of the sudden increase keep changing?

Page 14, line 3: " that increases their western horizontal ...."

Page 14, line 11: growth and intensification are two different things. So which one is it? Follows the thermal wind balance means that they are in geostrophic equilibrium, for most part based on Figure 8? But they seem to become more ageostrophic as they intensify, probably due to the effect of ageostrophic filaments.

Figure 10: maybe adding more isotherms would help relating the text to the figure.

Page 15, line 5: 50% of what. Maybe the sentence should be rephrased. Page 15, line 7: "Sea-surface salinity decreased in both .."

Page 15, line 8: "this freshening is related to the presence of a subsurface salinity maximum in the Caribbean Sea, causing upwelled waters to be more saline than surface waters." How does that make sense? Please rephrase. What is the name of the water mass that constitutes the salinity maximum?

Page 16, line 14: what is "mesoscale variability"? It doesn't mean anything in the context of this sentence. Are the authors talking about a meridional average, of the maximum along each meridian line?

Page 16, line 17:" the EKE increased by 123% ... "

Page 16, line 18:" and Ekman75 resulted in ..."

Page 16, line 32: "even though they are only due to 30-40% of the total number of anticyclones in this region", meaning they constitute only 30-40% of the total number
of ACs?

Page 18, line 7: based on what numbers of figures do you make this statement? The standard deviation in Table 1?

Page 18, line 8: "Similar observations by (Centurioni and Niiler, 2003), we ...."

Page 18, line 11: Is this something observed in real data? How much cyclones contribute to EKE? And why less cyclones with stronger upwelling?

Page 18, line 17: "These simulations have relatively lower vertical shear at 65oW than ???. "

Page 21, line 19: cite Table 1 at the end of the sentence.

Page 22, line 7: "Furthermore, we showed how the westward intensification of Caribbean anticyclones could be driven by baroclinic instabilities". This was not shown is this study. Please remove statement.

Page 22, line 17: the authors previously stated that both salinity fluxes and wind have to be accounted for to explain the variability. So how reliable is the relationship with the wind only?

Page 22, line 18-19: the authors are saying that the variance is higher, but it was previously shown that there was less eddies. So what causes the higher variance?

Page 22, last line: "processes explain some of the mesoscale variability in the Caribbean Sea"

---

## Author Comment (AC1) · 16 Sep 2019

**Reply to reviewer 1**

**Summary**

We thank reviewer 1 for the useful comments. We considered all comments and adjusted the manuscript. Line numbers correspond to the new manuscript.

**Major comment**

Previous studies suggested that enhancement of mesoscale anomalies in the Central Caribbean coincide with the presence of shallower bottom topography (p.e. Andrade and Barton, 2000) when reaching the Beata Ridge. Would the authors analyze that aspect and comment on that regard.

- Indeed, previous studies suggested several mechanisms that impact the life cycle of Caribbean anticyclones, among which topography. (Introduction: page 2, lines 6-10: *"They can be generated from the interaction of the flow with the topography (Molinari et al., 1981; Andrade and Barton, 2000), from the meandering current (Andrade and Barton, 2000), from instabilities due to the presence of the river plume (Chérubin and Richardson, 2007), and from perturbations caused by the interaction of North Brazil Current Rings (NBC Rings) with topography (e.g., Simmons and Nof, 2002; Goni and Johns, 2003; Jochumsen et al., 2010)"*).

  While this is an interesting topic, it is not the main topic of study in this manuscript: here we analyse how the wind-driven upwelling affects the westward intensification. (Page 2, lines 32-33: *"In this study, we hypothesize that this intensification is steered by the offshore advection of cold upwelling filaments that cool the interior of the basin."*).

  We did not explicitly study the impact of topography. However, in the revised version of Section 4, we now also discuss the variations in westward intensification that occur with longitude (Fig. 8d). In this plot, we identify 3 longitudes at which the intensification peaks. Here, we discuss a possible connection to the local topography:

  **Page 13, lines 6-7:**
  *"Although the third rapid increase is located eastward of the Beata Ridge at 73ºW, it is possible that this topographic feature has some impact on the westward intensification, as was previously proposed by Andrade and Barton (2000)."*

**Minor comments**

1. Page 2, line 19

"Here , model studies have shown that Caribbean anticyclones could influence eddy-shedding events of the Loop Current (Oey et al., 2003; Murphy et al., 1999; Carton and Chao, 1999; Candela, 2003; van 20 Westen et al., 2018)."

- Corrected (Page 2, line 19)

2. Page 5, line 1-2

"We will use this set of simulations  to study aspects of the seasonal and interannual variability of Caribbean anticyclones."

- Sentence removed.

3. Page 6, line 11

*"The considered region was limited to 1.5 × R_{eddy}. This restriction was applied to ensure that the advection of cold filaments and other mesoscale variability was excluded from the analysis."*

Why excluded?

- The goal of this study is to analyse the westward intensification of Caribbean anticyclones. Therefore, we wanted to isolate this westward intensification from other processes that contribute to the eddy kinetic energy. We rewrote this paragraph to clarify:

  **Page 6, lines 17-21:**
  *"Since the focus of this study is to analyze the westward intensification of the anticyclones, we not only calculate the EKE and strength of the horizontal density gradients over the full domain, but we also calculate their contribution associated with the anticyclones only. We estimate the latter by considering the EKE and density gradients around the core of an eddy at each 5-day averaged field over a spatial extend of 1.5 × R_{eddy}. This procedure allows us to study only the westward intensification of the anticyclones."*

4. Page 6, lines 23, 25, 26 and 27

Modify Andrade (2003) for Andrade et al. (2003): Andrade, C.A., E.D. Barton and C.N.K. Mooers, Evidence for an Eastward Flow along the Central and South American Caribbean Coast, Journal of Geophysical Research, Vol. 108, C6-3185, June, 2003. EID:2-s2.0-0141501411.

- The reference was replaced.

5. Page 7, lines 8-11

*"The Guajira upwelling region is located west of the Cariaco upwelling region, between 69°W and 74°W (Rueda-Roa and Muller-Karger, 2013). Here, the observed average SST is slightly higher (25.5°C) than in the Cariaco upwelling region (Rueda-Roa and Muller-Karger, 2013). The model displays a similar temperature difference between the two upwelling regions (26.1°C in Guajira, Fig. 3d)."*

Consider compare these temperature values with those in Andrade and Barton (2005).

- We made a comparison with Andrade and Barton (2005) and added a sentence to the manuscript.

  **Page 8, lines 2-3**
  *"Furthermore, the modeled temperatures are in line with Andrade and Barton (2005), who found surface temperatures varying between 25.6°C and 28°C in the Guajira upwelling region."*

6. Page 8, lines 4-8

*"In line with observations (Richardson, 2005; Carton and Chao, 1999), we find that the flow in the Caribbean Sea is highly variable (Fig. 4). In the eastern part of the basin, the surface EKE is relatively low (100-300 cm² s⁻², Fig 4a). The EKE increases westward towards a maximum >900 cm² s⁻² at 78°W."*

Consider compare these EKE values with those in Andrade and Barton (2000) in this sentence and in the other parts where EKE was commented throughout the manuscript.

- We compared the average EKE in Venezuela Basin with Andrade and Barton (2000), and added a sentence about this.

  **Page 8, lines 16-27**

*"In line with observations (Andrade and Barton, 2000; Richardson, 2005; Carton and Chao, 1999), we find that the flow in the Caribbean Sea is highly variable (Fig. 4). In the eastern part of the basin, the modeled surface EKE is relatively low (100-300 $cm^2 s^{-2}$, Fig 4a) and similar to observations of Andrade and Barton (2000). The EKE increases westward towards a maximum >900 $cm^2 s^{-2}$ at 78oW. The modeled magnitude of EKE is higher than found in satellite altimetry (>600 $cm^2 s^{-2}$, Andrade and Barton, 2000; Jouanno et al., 2012), but it is more similar to estimates obtained from surface drifters (>900 $cm^2 s^{-2}$, Richardson, 2005). This is in line with other modeling studies (Jouanno et al., 2008, 2012), and this discrepancy is mainly attributed to the coarse resolution (0.25o) of the gridded altimetry data products (Jouanno et al., 2008). The modeled spatial variability of EKE matches analyses of to satellite altimetry well (Jouanno et al., 2012; Ducet et al., 2000). Corresponding to observations (Silander, 2005; van der Boog et al., 2019), we find that the eddy kinetic energy is surface intensified (Fig. 4b). In line with the modeling results of Jouanno et al. (2008), the magnitude of EKE at depth also increases towards the west (Fig. 4b)."*

- Furthermore, we compared Fig. 11 of Andrade and Barton (2000) to the meridional increase of EKE in Ekman100.

  **Page 11, line 12; Page 12, lines 1-2**
  *"Overall, the meridional maximum of EKE that is contained in the anticyclones increases from approximately 200 $cm^2 s^{-2}$ at 65oW towards 530 $cm^2 s^{-2}$ at 75oW (Fig. 8a). These values are similar to those observed in Andrade and Barton (2000)."*

7. Page 9, line 13
*"The cyclones are less energetic than the anticyclones."*
This is not true in the southwest Caribbean where cyclonic circulation is almost permanent. Consider complete the sentence with a location reference
- We clarified the sentence by adding a location.

  **Page 10, lines 11-12**
  *"In the central Caribbean Sea (65oW-75oW and 12.5oN-17.5oN), the cyclones are less energetic than the anticyclones, and have an average amplitude of $A_{eddy} = -0.16$ m and swirl velocity of $u_{swirl} = 0.50$ m $s^{-1}$."*

8. Page 22, line 21
*"...that  observed salinity in the Cariaco Basin was anomalously low in these year:"*
- Corrected (Page 23, line 25).

9. Page 22, line 33
*"Together these two processes explain the mesoscale variability in the Caribbean Sea."*
Clarify why the wind stress field is not included in this sentence.
- The wind stress field is similar in each simulation, only the magnitude differed. Therefore, the impact of the wind stress field (spatial variations) should not differ between the simulations, and these were not analysed. To highlight this, we clarified the first sentence of the paragraph.

  **Page 24, lines 1-2:**
  *"Overall, in this study we showed how the strength of the zonal wind stress in the Caribbean Sea impact the eddy variability in this basin."*

10. Page 22, line 26
*"(Villamizar G. and Cervigón, 2017), it also impacts the mesoscale variability."*
no G.in the reference
- Corrected. (Page 23, line 31)

11. Page 23, line 31
Is Juan Manuael Sayol, Y pma, is Ypma
- Corrected (Page 25, line 4).

---

## Author Comment (AC2) · 16 Sep 2019

**Reply to reviewer 2**

We thank reviewer 2 for the comments. In particular, we liked the comments concerning Section 4 and used those to rewrite this section and added panels to Figure 8. We think this clarified our story substantially. Line numbers correspond to the new manuscript.

**Summary of reviewer**

In this manuscript the authors use a numerical simulation to study the effect of up-welling filament advected by anticyclonic eddies in the eastern Caribbean basin. They showed that upwelling filaments, mostly from the upwelling centers are entrained on the western side of the eddies and contribute to their westward intensification. The intensification gets stronger when the upwelling is stronger and vice and versa but it can be influenced by the low salinity of the Amazon and Orinoco plume, which is also influenced by the wind. I think that this study in a good contribution to the understanding of the dynamics of the circulation in the Caribbean Sea and the authors have made an excellent job at describing the mechanics of the intensification process within the case studies that they analyzed. But it would have been good to show in a simulation with realistic forcing that reproduce the seasonal cycle, that ACs intensifies as they propagate westward.

▪ To isolate the impact of upwelling on the life cycle of the anticyclones, we removed all seasonality from the model. We added a sentence to clarify this in the introduction and clarified the model configuration section.

**Page 3, lines 3-5**
*"To isolate the impact of upwelling on the life cycle of the anticyclones, we kept all other forcing parameters constant."*

**Page 3, line 27**
*"All simulations were forced with time-averaged surface and lateral boundary conditions."*

**Page 4, lines 3-4**
*"The reference simulation (Ekman100) is forced by a time-averaged zonal wind forcing, computed from the wind fields of years 2007-2017 of ERA-Interim (Dee et al., 2011)."*

▪ The variations in wind forcing between the runs cover the seasonal variability of the wind strength. This allows us to speculate on some aspects of the seasonal cycle, but has some limitations, because the wind forcing and for example, the river plumes have a different seasonality. We agree that this is an interesting point, and suggest that this could be the focus of a future study. We clarified this in the discussion.

**Page 23, lines 15-17**
*"However, the dispersal of the Amazon and Orinoco River plumes also has a distinct seasonal cycle (Hellweger and Gordon, 2002; Chérubin and Richardson, 2007), which is absent in our model and would be an interesting topic for a further study."*

**Minor comments**

**Abstract edits**

1. Page 1, lines 4-5
*"These dense filaments are advected by the anticyclones, leading to an increase of the horizontal density gradients on the western side of the anticyclones".*

Should read "Following the thermal wind balance, the increased density gradients result in an increase of the vertical shear of the anticyclones and of their westward intensification."
  Corrected (Page 1, lines 4-5).

2. Page 1, lines 8-10
*"As expected, stronger (weaker) upwelling is associated with  stronger (weaker) offshore cooling and a more (less) westward"*
Remove "more".
  ▪ Corrected (Page 1, lines 7-8).

3. Page 1, lines 11-13
Last sentence is difficult to understand. Should be rephrased. Especially without the context of the study. Think of someone who'd just reading the abstract, it would be unable to follow.
  ▪ We agree and we changed the last sentence.

   **Page 1, lines 8-14**
   *"Moreover, the simulations with weaker upwelling show farther advection of the Amazon and Orinoco River plumes into the basin. As a result, in these simulations the horizontal density gradients were predominantly set by horizontal salinity gradients. The importance of the horizontal density gradients driven by temperature, which are associated with the upwelling, increased with increasing upwelling strength. The results of this study highlight that both upwelling and the advection of the river plumes affect the life cycle of mesoscale eddies in the Caribbean Sea."*

Manuscript edits
4. Page 2, lines 12-13
*" and propagate along the wind-driven upwelling regions along the South-American coast".*
Remove the "along" repetition.
  ▪ Corrected (Page 2, lines 13-14).

5. Page 3, lines 10-20
What topography was used and how the passages between the island were accounted for in the 1/12-degree resolution grid? Was the transport between the island estimated and compared to observations? How did the model perform in the other passages?
  ▪ We added information about the topography in Section 2.1.

   **Page 3, lines 15-18**
   *"The same topography as the Operational Mercator global ocean analysis (Mercator) of the E.U. Copernicus Marine Service Information was used, where the topography is based on ETOPO1 for the deep ocean and near the coast on GEBCO8 for the coast and slopes. In this topographic setup, the majority of the islands of the Lesser Antilles are represented (Fig. 1). The islands that are too small to be captured are modeled as shallow ridges."*

  ▪ We also added a brief discussion on the transport in Section 3.1.

   **Page 6, lines 25-28**
   *"The flow through the Lesser Antilles is highly variable and depends on variations of the flow upstream and on the wind forcing (Johns et al., 2002). Based on scarce data, Johns et al. (2002) estimated the transport at 66°W at 18.4 ± 4.7 Sv. In our simulation, the transport is more stable due to the stationary forcing and is on*

*average 13.6 Sv at 66°W, which is close to the lower range of the estimate of Johns et al. (2002)."*

6. Page 3, line 20-25
It is difficult to understand how the forcing were applied. It is said that stationary conditions were applied, but it is not clear exactly which ones. For instance, the SST relaxation was not stationary? Was it released to monthly averaged SST? What about heat and freshwater fluxes?

▪ All forcing fields were applied year-averaged forcing. We clarified this in the text.

**Page 3, lines 3-5**
*"To isolate the impact of upwelling on the life cycle of the anticyclones, we kept all other forcing parameters constant."*

**Page 3, line 27**
*"All simulations were forced with stationary surface and lateral boundary conditions."*

**Page 4, lines 1-2**
*"The Orinoco River, Magdalena River and Mississippi River are prescribed as stationary fresh water fluxes at the open boundaries with discharges based on Fekete et al. (2000)."*

**Page 4, lines 3-4**
*"The reference simulation (Ekman100) is forced by a time-averaged zonal wind forcing, computed from the wind fields of years 2007-2017 of ERA-Interim (Dee et al., 2011)."*

7. Page 3, last paragraph
Did all simulations used stationary boundary conditions? Not only Ekman 100, right?

▪ Yes, all simulations used stationary boundary conditions. We clarified this in the text.

**Page 3, line 27**
*All simulations were forced with stationary surface and lateral boundary conditions.*

**Page 4, lines 3-4**
*"The reference simulation (Ekman100) is forced by a time-averaged zonal wind forcing, computed from the wind fields of years 2007-2017 of ERA-Interim (Dee et al., 2011)."*

8. Page 3, line 30:
I don't understand the sentence: "*the upwelling regions corresponds to of the year-averaged northward Ekman transport (100%),"*. How is the year averaged calculated? Is it over the 20-year simulation? But which simulation since in your case studies the simulations cannot be realistic because of the stationary boundary conditions?

▪ We agree that this sentence is confusing. We clarified this paragraph, for more details see reply to Comments 6 and 7.

9. Page 4, Line 5:
"*with a constant proportional to the wind stress at the upwelling regions".*

Not sure the sentence expresses what the authors are trying to say. The authors mean to talk about the wind stress magnitude that was reduced by the same amount between simulations. Please revise.

- We revised the paragraph.

    **Page 4, lines 9-16**
    *"To investigate the effect of wind-driven upwelling on the westward intensification of anticyclones, only this zonal wind forcing ($\tau_x$) was altered between simulations. The magnitude of the zonal wind stress was reduced or increased in each simulation by the same constant over the entire domain (Fig. 2). With this approach, we ensured that we only change the upwelling strength and not the curl of the wind stress. The magnitude of the reduction or increase was determined based on the zonal wind stress in the upwelling region in Ekman100. In Ekman150, the zonal wind stress in the upwelling region was 50% stronger than the wind stress in Ekman100, resulting in a theoretical increase of the northward Ekman transport of 50%. The wind stress along the coast was weaker in the Ekman50, leading to a theoretical weaker upwelling (50%) in this simulation compared to Ekman100. The same principle was applied in Ekman75 and Ekman125."*

10. Page 4, line 14
Please indicate the figure that shows the upwelling centers location.

- Corrected. (Page 5, line 2)

11. Page 5, first line
*"We will use this set of simulations with the different to study aspects of the seasonal and".*
Please revise.

- Sentence deleted.

12. Page 5, lines 14-18
*"the swirl velocity as the average of the maximum northward and maximum southward velocity of the eddy".*
Is the location estimated from the center of the SSH anomaly or the point of maximum SHH? Also, why using the meridional velocity only? Why not the location of the maximum of the radial velocity instead?

- The location of the center was estimated using the py-eddy tracker. The eddy tracker already selects the location of maximum SLA, and therefore we used that location. We clarified this in the text.

    **Page 5, lines 19-20**
    *"At the center of the eddies provided by the eddy tracker, we extracted the amplitude ($A_{eddy}$), swirl velocity ($u_{swirl}$), radius ($R_{eddy}$) and properties from the model output to assess their characteristics."*

- Since the py-eddy tracker identifies circular sea-level anomalies as eddies, we assumed that the shape of the eddies was circular. This allowed us to estimate the swirl velocity of the eddies from the meridional velocities. We averaged both the northward and southward maximum velocity to obtain a more robust estimate of the eddy radius. We clarified this assumption in the text.

    **Page 5, line 21; page 6, lines 1-3**
    *"Since the py-eddy tracker identifies circular anomalies as eddies, we could define the swirl velocity as the average of the maximum northward and maximum*

*southward velocity of the eddy. The location of both velocities was used to obtain a robust estimate of the eddy radius ($R_{eddy}$), which was defined as half the distance between these locations."*

13. Page 6, line 3
The thermal wind balance is not a force or a driver, but rather another way to express the geostrophic balance. So, it is meaningless to say how it affected the westward intensification. It is simply the horizontal density gradient as you express it. It can be related to the vertical shear through the thermal wind balance equations.
  ▪ We agree with the reviewer and changed the sentence.

  **Page 6, line 9-10**
  *"To gain insight in the geostrophic part of the westward intensification of the anticyclones in each simulation, we computed the strength of the horizontal density gradients ($|\nabla\sigma|$)."*

14. Pages 6, line 9:
Can you show how sigma_T and sigma_S contribute to sigma? What equation of state was used?
  ▪ We used a linear equation of state. Because we only used temperature and salinity differences, there was no need for a reference temperature or salinity.

  **Page 6, lines 13-16**
  *"The contribution of temperature ($|\nabla\sigma_T|$) and salinity ($|\nabla\sigma_S|$) to the horizontal density gradients was computed in a similar manner, where the density differences were calculated from a linear equation of state as $\Delta\sigma T = \rho_0\alpha(T_1 - T_0)$ and $\Delta\sigma S = \rho_0\beta(S_1 - S_0)$, respectively. Here, $\alpha$, $\beta$ and $\rho_0$ are constants; $\alpha = -3.1 \times 10^{-4}\ ^oC^{-1}$, $\beta = 7.2 \times 10^{-4}\ psu^{-1}$, and $\rho_0 = 999.8\ kg\ m^{-3}$."*

15. Page 6, line 18
Should read "*Further west,...at 17N..*" maybe the longitude can begiven here as well because the sentence starts with "Further west...".
  ▪ We added a longitude.

  **Page 6, lines 28-30**
  *"Further westward at $80^oW$, the modeled flow accelerates over shallow topography at $17^oN$, where it continues northwestward towards Yucatan Channel into the Gulf of Mexico."*

16. Page 7, line 11
I think the authors meant Fig. 3c.
  ▪ That is indeed correct: we changed Fig 3d to Fig. 3c. (Page 8, line 2)

17. Page 7, lines 12-15
Over what depth this density gradient can be observed. I would imagine that it strongly depends on the thickness of the fresh water plume? All the dynamics discussed in this study is limited to the first 50 meters, which is the vertical extension of most eddies. What happens below? Do the surface eddies have a deep signature and are they also intensified at depth?
  ▪ In Ekman100, the average depth of the pycnocline in the Caribbean Sea is located at approximately 50 m depth. We added a sentence to the text to highlight this.

Furthermore, as visible in Figure 4, the eddy kinetic energy is surface intensified. This is in line with observations of Silander (2005) and van der Boog et al. (2019).

**Page 6, lines 2-4**
*"To characterize the local properties (T,S,σ) of the background and the eddies, these variables were averaged over the upper 50m. This depth corresponds to the average 5 depth of the pycnocline in the Caribbean Sea. The latter ensures that these properties not only reflect variations in surface forcing."*

**Page 7, lines 9-11**
*"Ekman100 displays a strong meridional density gradient in the mixed layer that varies between $\sigma = 25.1$ kg m$^{-3}$ in the south (11ºN) and $\sigma = 22.7$ kg m$^{-3}$ in the north (18ºN, Fig. 3b). The strongest meridional gradients are close to the surface and co-located with the Caribbean Current."*

**Page 8, lines 8-9**
*"The magnitude of the zonal density gradient is largest in the surface mixed layer and decreases rapidly below."*

**Page 8, lines 24-26**
*"Corresponding to observations (Silander, 2005; van der Boog et al., 2019), we find that the eddy kinetic energy is surface intensified (Fig. 4b). In line with the modeling results of Jouanno et al. (2008), the magnitude of EKE at depth increases towards the west (Fig. 4b)."*

18. Page 8, lines 10
"corresponds" could be replaced by "matches".
- Corrected. (Page 8, line 24)

19. Figure 5, caption:
*"Near-surface properties of the Caribbean Sea, averaged over 5 days in Ekman100 in year 20"*
- We clarified the caption of this figure.

  **Figure 5, caption**
  *"Near-surface properties of the Caribbean Sea, averaged over a 5-day period during the final year of the simulation Ekman100."*

20. Page 9, line 17
Not sure the first sentence is correct here. None of what it says has been proven yet.
- Agreed. We rephrased the sentence.

  **Page 10, lines 18-19**
  *"There is a strong and significant correlation between the amplitude of the tracked mesoscale eddies and the surface EKE. This suggests that the westward increase of EKE is related to the strength of the eddies."*

21. Page 10, line 6
Do the authors mean *"To assess the contribution of the anticyclones with the long tracks to the total EKE variability..."*?
- Yes, we did meant to say that. We changed the sentence.

**Page 10, lines 26-28**

*"To assess the contribution of the anticyclones with the long tracks **to** the total EKE variability, we calculated their EKE from the zonal and meridional velocity fields by taking into account the EKE within 1.5 × $R_{eddy}$ of each eddy as described in Section 2."*

22. Page 11 line: which component of the velocity is used? Only v or the magnitude?
   ▪ We used both u and v. We clarified this in the text.

**Page 10, lines 26-28**

*"To assess the contribution of the anticyclones with the long tracks to the total EKE variability, we calculated their EKE **from the zonal and meridional velocity fields** by taking into account the EKE within 1.5 × $R_{eddy}$ of each eddy as described in Section 2.*

23. Page 11, line 9:

*"At 64°W and 71°W, the vertical shear of the anticyclones increases zonally more rapidly."*
I have a problem with this statement. It is based on visual assessment, which is difficult to prove. Figure 8 could show that, but it only starts at 65W, so one can't see the strong increase. Also, the strong increase at 71°W is not visible. Then it becomes harder to see the link with the upwelling centers, although I think the link between the sharp shear increase and the upwelling filament average position is a viable argument. Maybe Figure 8 can be expanded to show that?
   - We expanded Figure 8 with an extra panel (panel d) to show the zonal increase of the properties. We also added lines at the maximum increase to clarify the regions of more rapid intensification.

**Figure 8**

[Figure]

**24. Page 11, line 10**
*"A comparison to the average shear of the total velocity field indicates that these longitudes are located close to two regions with strong background vertical shear (black contour in Fig. 7a)."*
I don't understand what is done here.
- For clarity, we removed this sentence.

**25. Page 11, lines 15-16**
What else than temperature and salinity the density gradient could be due to?
- Agreed. We deleted the sentence to clarify the paragraph.

**26. Page 11, line 21:** to make such statement, which is not obvious in Figure 8, the slopes along the curve could be shown on Figure 8
- We expanded Figure 8 with an extra panel to show the zonal increase of the properties. Furthermore, we added a new paragraph to describe this new panel.

   **Page 12, lines 13-14; Page 13, lines 1-9**
   *"Because the westward increase of the horizontal density gradients and vertical shear of the anticyclones (Fig. 8b) is not constant, we computed the variations with longitude in westward direction of these quantities (Fig. 8d). From this, three regions can be identified as locations of more rapid intensification (64.6°W, 66.7°W, 72°W). Up to 64.6°W, the westward increase of horizontal density gradients is not fully balanced by the vertical shear, indicating that the westward intensification of the anticyclones is not in geostrophic balance at this stage. The westward intensification becomes more geostrophic towards the second peak of rapid intensification at 66.7°W (Fig. 8d). This peak is located near the Cariaco upwelling region. As the anticyclones move closer towards the Guajira upwelling region, they intensify more rapidly again (at 72°W in Fig. 8d). Although the third rapid increase is located eastward of the Beata Ridge at 73°W, it is possible that this topographic feature has some impact on the westward intensification, which was previously proposed by Andrade and Barton (2000). However, our model result suggest that all three regions of more rapid increase are located close to preferred locations of the shedding of upwelling filaments (Fig. 5c)."*

**27. Page 11, line 26-27:** it means that AC are not fully geostrophic.
- We added a sentence to clarify this.

   **Page 12, lines 4-5**
   *"In line with the low Rossby number of the anticyclones, there is a small difference in magnitude between the vertical shear and density gradients indicating the presence of small ageostrophic velocities."*

**28. Page 11, line 31:** cite Figure 7(c&d) after *"...differences."*
- Corrected. (Page 12, line 5)

**29. Figure 8:**
The average AC density anomaly could also be shown and the figure should start at 64W.
- We added a panel to Figure 8 (panel c) to show the average density anomaly of the background and of the anticyclones. Furthermore, we expanded Figure 8 towards the east (62.5°W). For consistency, we also expanded the longitudes in Figures 12 and 13. See comment 23 for the updated Figure 8.

**30. Page 13, line 9**
Why is the location of the sudden increase keep changing?
- We agree that this section was unclear, and rewrote the last paragraphs of Section 4 to clarify the sudden increase. See Comment 26 for the altered paragraph.

**31. Page 14, line 3**
*" that increases their western horizontal…"*
  ▪ Corrected (Page 14, line 31-32)

**32. Page 14, line 11**
Growth and intensification are two different things. So which one is it? Follows the thermal wind balance means that they are in geostrophic equilibrium, for most part based on Figure 8? But they seem to become more ageostrophic as they intensify, probably due to the effect of ageostrophic filaments.
  ▪ We mean intensification. For consistency, we replaced everywhere growth by intensification everywhere. We also rewrote section 4 to comment more on the ageostrophic and geostrophic intensification of the anticyclones. See comment 26 for the new paragraph.

**33. Figure 10**
Maybe adding more isotherms would help relating the text to the figure.
  ▪ Corrected. We also altered the colors of the contour lines for clarity.

**Figure 10**

[Figure]

**34. Page 15, line 5**
50% of what. Maybe the sentence should be rephrased.
  ▪ We rephrased the sentence to clarify.

  **Page 15, lines 12-13**
  *"The zonal wind stress in Ekman50 and Ekman75 was reduced compared to Ekman100, such that the wind stress in the upwelling region was 50% and 25% less than Ekman100, respectively."*

**35. Page 15, line7**

*"Sea-surface salinity decreased in both"*
- Corrected (Page 16, line 2).

36. Page 15, line 8
*"this freshening is related to the presence of a subsurface salinity maximum in the Caribbean Sea, causing upwelled waters to be more saline than surface waters."*
How does that make sense? Please rephrase. What is the name of the water mass that constitutes the salinity maximum?
- We added the name of the water mass and replaced the sentence.

   **Page 16, lines 3-5**
   *"This freshening is related to the presence of a subsurface salinity maximum in the Caribbean Sea due to the presence of a water mass, referred to as Subtropical Underwater. This water mass is located at approximately 100 m depth and leads to more saline upwelled waters compared to the fresher surface waters (van der Boog et al., 2019)."*

37. Page 16, line 14:
What is *"mesoscale variability"*? It doesn't mean anything in the context of this sentence. Are the authors talking about a meridional average, of the maximum along each meridian line?
- We clarified this in the text.

   **Page 18, lines 1-2**
   *To gain insight into the westward increase of EKE, the maximum of the total EKE between 12.5°N and 17.5°N was computed as a function of longitude (Fig. 12a).*

38. Page 16, line 17:
*" the EKE increased by 123%..."*
- Corrected (Page 18, line 3).

39. Page 16, line 18
*" and Ekman75 resulted in ..."*
- Corrected (Page 18, line 4)

40. Page 16, line 32:
*"even though they are only due to 30-40% of the total number of anticyclones in this region"*, meaning they constitute only 30-40% of the total number of ACs?
- We clarified this in the text.

   **Page 18, lines 18-20**
   *"In the simulations with stronger upwelling (Ekman100, Ekman125 and Ekman150), the anticyclones with long tracks (Fig. 12b) are responsible for more than half of the total EKE (Fig. 12a), even though they constitute only 34-43% of the total number of anticyclones in this region (Table 1)."*

41. Page 18, line 7
Based on what numbers or figures do you make this statement? The standard deviation in Table 1?
- Yes, we based this the standard deviation in Table 1. We clarified this in the text.

   **Page 19, lines 5-7**
   *"It is interesting to note that the variation of the properties of the anticyclones is very*

*small in each simulation (standard deviation of the anticyclone properties in Table 1)."*

**42. Page 18, line 8**
*"Similar observations by (Centurioni and Niiler, 2003), we.."*
- Changed it. (Page 19, line 10)

**43. Page 18, line 11**
Is this something observed in real data? How much cyclones contribute to EKE? And why less cyclones with stronger upwelling?
- Observations of the cyclones are limited, so it is unknown whether this is observed in real data. The cyclones contribute less to the total EKE than the anticyclones (see Figure 6c-d for Ekman100). Furthermore, the variations in cyclone formation rate between the simulations was not significant.

    **Page 19, lines 10-17**
    *"In line with observations of Centurioni and Niiler (2003), we found less cyclones than anticyclones in each simulation (Table 1). The lowest average formation rate of cyclones is found in Ekman150 with 4.15 cyclones per year, and the highest average formation rate is present in Ekman50 with 6.15 cyclones per year. These formation rates were highly variable, and differences in formation rates between simulations were not significant. In none of the simulations, the py-eddy tracker was able to track multiple cyclones from east to west (65ᵒW-75ᵒW). This implies that the cyclones are either deformed or dissipated too much, such that the py-eddy tracker could not track their sea-level anomaly. Overall, the behavior of the mesoscale eddies is similar in all simulations and the spatial pattern and magnitude of the surface EKE is governed by the anticyclones with long tracks."*

**44. Page 18, line 17**
*"These simulations have relatively lower vertical shear at 65ᵒW than???. "*
- We changed the sentence to clarify the comparison.

    **Page 22, lines 3-4**
    *"In the interior of the basin, the horizontal density gradients strengthen in Ekman50 compared to Ekman100 (Fig. 14a), while they are weaker in Ekman150 compared to Ekman100 (Fig. 14c)."*

**45. Page 21, line 19**
Cite Table 1 at the end of the sentence.
- Done. (Page 22, line 24)

**46. Page 22, line 7**
*"Furthermore, we showed how the westward intensification of Caribbean anticyclones could be driven by baroclinic instabilities".*
This was not shown is this study. Please remove statement.
- Agreed. Statement is removed.

**47. Page 22, line 17**
The authors previously stated that both salinity fluxes and wind have to be accounted for to explain the variability. So how reliable is the relationship with the wind only?
- We rewrote the paragraph to address this comment.

**Page 23, lines 18-28**
*"The results of this study also highlight some aspects of the interannual variability of the eddy field in the Caribbean Sea. In our simulations, stronger wind forcing resulted in a higher EKE in the center of the Caribbean Sea. Assuming that this relation holds on interannual time scales as well, these results suggest a positive correlation between the wind forcing and eddy variability in the interior of the basin. Jouanno and Sheinbaum (2013) used a model with seasonally varying boundary conditions and identified a similar relationship. This is also found in observations (Fig. 15), which show that sea-surface variance is higher in years with stronger zonal winds. Figure 15 also suggests that the response of the sea-surface variance to the wind stress is non linear: Although 2010 and 2011 were years with weak zonal winds, the sea-surface variance was relatively high. It is interesting to note that the salinity in the Cariaco Basin was anomalously low in these years (Cariaco project, 2019). Taking into account the shallow depth of this salinity anomaly, it is plausible that it is related to the farther westward propagation of the river plumes as seen in Ekman50. This supports our view that both upwelling and the dispersal of the river plumes affect the mesoscale eddy field in the Caribbean Sea."*

48. Page 22, line 18-19
The authors are saying that the variance is higher, but it was previously shown that there was less eddies. So, what causes the higher variance?
- We added a sentence to the paragraph to clarify the origin of the higher variance.

**Page 23, lines 18-28**
*"In our simulations, stronger wind forcing resulted in a higher EKE in the center of the Caribbean Sea. […] This supports our view that both upwelling and the dispersal of the river plumes affect the mesoscale eddy field in the Caribbean Sea."*

49. Page 22, last line
*"processes explain some of the mesoscale variability in the Caribbean Sea"*
- Corrected (Page 24, lines 4-5).